# Text to Sketch Generation with Multi-Styles

**Tengjie Li** [1], **Shikui Tu**[1*], **Lei Xu**[1,2] *

[1]School of Computer Science, Shanghai Jiao Tong University

[2] Guangdong Laboratory of Artificial Intelligence and Digital Economy (SZ), Guangdong, China

{765127364, tushikui, leixu}@sjtu.edu.cn

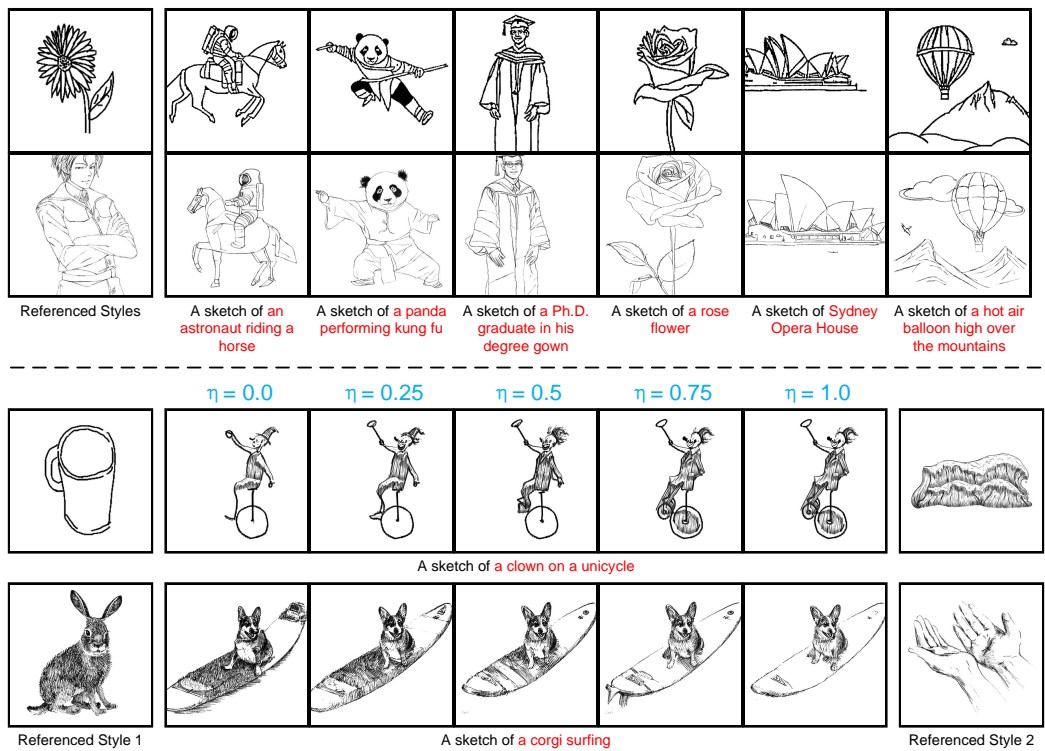

Figure 1: **Top**: Synthesized sketches from a specific-style exemplar by our proposed method. **Bottom**: Multi-style sketches generated by our framework. $\eta$ is used to control style tendency. As $\eta$ increases, the result's style becomes more aligned with the referenced style 2, and vice versa.

## Abstract

Recent advances in vision-language models have facilitated progress in sketch generation. However, existing specialized methods primarily focus on generic synthesis and lack mechanisms for precise control over sketch styles. In this work, we propose a training-free framework based on diffusion models that enables explicit style guidance via textual prompts and referenced style sketches. Unlike previous style transfer methods that overwrite key and value matrices in self-attention, we incorporate the reference features as auxiliary information with linear smoothing and leverage a style-content guidance mechanism. This design

---

*Correspondence authors are Shikui Tu and Lei Xu.

39th Conference on Neural Information Processing Systems (NeurIPS 2025).

effectively reduces content leakage from reference sketches and enhances synthesis quality, especially in cases with low structural similarity between reference and target sketches. Furthermore, we extend our framework to support controllable multi-style generation by integrating features from multiple reference sketches, coordinated via a joint AdaIN module. Extensive experiments demonstrate that our approach achieves high-quality sketch generation with accurate style alignment and improved flexibility in style control. The official implementation of M3S is available at `https://github.com/CMACH508/M3S`.

# 1   Introduction

Sketching, as a universal visual medium with historical roots spanning millennia, demonstrates remarkable accessibility by transcending age and cultural barriers [54]. This unique capacity to convey complex concepts through minimal strokes enables effective cross-linguistic communication, establishing it as a potent tool for ideation and conceptualization. Through historical development, sketching has evolved into a multidisciplinary practice permeating diverse domains ranging from industrial prototyping and artistic expression to educational visualization and recreational applications. This stylistic diversity, amplified by modern digital tools, introduces significant challenges for automated sketch generation systems to capture and reproduce specific artistic styles accurately.

A fundamental challenge in advancing sketch generation methods originates from the inherent difficulties in data acquisition. Unlike natural images that can be readily obtained through web scraping, high-quality sketch datasets necessitate specialized artistic expertise and substantial time investment for creation, resulting in constrained dataset scales [37, 50, 57]. While freehand sketches with rough strokes exhibit greater accessibility [7, 28, 35], including large-scale collections like QuickDraw [11], their abstract and sparse representations pose inherent limitations for training models to generate high-fidelity sketches with fine-grained details [4, 58]. Furthermore, this representational gap significantly impedes the effectiveness of captioning models [22] in producing accurate textual descriptions for sketches, thereby creating a critical bottleneck for text-conditioned sketch generation.

Recent work has explored leveraging pretrained vision-language models for sketch generation. CLIPasso [46] utilizes CLIP's cross-modal embedding space [32] to optimize Bézier curve parameters by minimizing a semantic consistency loss between rasterized sketches and natural images. Building on this idea, CLIPascene [45] introduces explicit scene decomposition to separate foreground and background elements. However, both methods depend heavily on existing image references for supervision. DiffSketcher [53] takes a step further by incorporating text-driven guidance. It combines Stable Diffusion [33] with Bézier curve optimization through score distillation sampling, enabling free-form sketch synthesis from textual prompts. Despite this advancement, existing methods [53, 18] suffer from limited control over stylistic attributes. In particular, text-based style conditioning often lacks the expressiveness and specificity needed to capture fine-grained visual traits, making it difficult to match exemplar styles precisely.

In exemplar-based style transfer, a prominent technical direction involves injecting visual features through the diffusion model's self-attention mechanisms. Representative works [3, 1, 5] propose swapping the $K/V$ matrices derived from the reference images' denoising process into target generation. However, this direct replacement strategy proves suboptimal when handling cross-domain scenarios: The inherent discrepancy between reference and target domains induces misalignment between generated queries $Q$ and substituted $K/V$ features [1], leading to both content leakage and deteriorated generation quality. [63] addresses this by introducing attention distillation to align target $K/V$ features with reference characteristics. However, this approach still suffers from excessive feature alignment that compromises output authenticity.

We propose **M**ulti-**S**tyle **S**ketch **S**ynthesis (M3S), a training-free framework for generating sketches with diverse and controllable styles. To balance stylistic fidelity and content preservation, we introduce a $K/V$ injection scheme composed of three components: (1) Hybrid attention fusion, which injects reference style features ($K_{ref}/V_{ref}$) with target features ($K_{tar}/V_{tar}$) in self-attention layers to integrate style while retaining content semantics; (2) Linear feature blending to mitigate content leakage from the reference; and (3) Separated guidance control, which divides classifier-free sampling into style and content directions, allowing flexible trade-offs between expression and

structure. This coordinated design enables high-quality, style-specific synthesis, as illustrated in the upper portion of Fig. 1.

We further investigate the feasibility of multi-style sketch generation. Sketch stylistic expression fundamentally manifests through stroke geometry and sparse texture patterns, contrasting with the dense pixel representations of natural images. Capitalizing on this characteristic, we extend M3S's feature injection to multi-style synthesis by integrating $K/V$ features from diverse references. Additionally, drawing inspiration from [1, 15], we implement a joint AdaIN modulation on intermediate denoising steps to regulate stylistic dominance. This allows users to adjust the blending weights $\eta$ between styles, for example, controlling the clown texture density in the second last row of Fig. 1.

In summary, our main contributions are as follows: (1) We propose a training-free framework for multi-style sketch generation. Key and value features from the referenced style sketches are considered auxiliary information, combined with linear smoothing and a style-content guidance mechanism, enabling the balance between style consistency and fidelity. (2) We investigate the potential of leveraging pre-trained diffusion models' prior knowledge for sketch generation with mixed styles. By introducing a joint AdaIN modulation mechanism, our approach provides users with flexible control over style generation tendencies. (3) We implement our method on Stable Diffusion v1.5 [33] and SDXL [29], with extensive experiments demonstrating the method's effectiveness, outperforming state-of-the-art approaches.

## 2   Related Work

**Sketch Synthesis**   The field of sketch generation has evolved through distinct methodological paradigms. Early breakthroughs like SketchRNN [11] established sequential modeling using RNN-based encoder-decoders for single-category sketch synthesis. Subsequent work enhanced multi-category generation through structured representations: pixel-space modeling [60, 25, 41], graph-based architectures [43, 31, 59], and stroke-level analysis [24, 51]. Recent diffusion-based approaches [49, 6] improved output quality but remain constrained to coarse category-level control. Although SketchAgent [47] exploits the powerful text comprehension capability of Large Language Models (LLMs) to guide LLMs to generate sketches by using drawing rules and reference drawing sequences as prompts, the generalizability of such approaches still needs to be further explored. Parallel image-to-sketch extraction research progressed from CNN-based methods [17, 23] (requiring paired training data) to training-free techniques leveraging vision-language priors [46, 45, 53, 55]. While existing solutions address specific aspects of sketch generation, they exhibit critical limitations in simultaneous style-text controllability.

**Image Style Transfer**   Style transfer has evolved significantly since the seminal work of Gatys et al. [10], who first demonstrated neural style transfer using Gram matrix-based feature statistics from pre-trained CNNs. Subsequent approaches improved efficiency by replacing iterative optimization with feed-forward networks [19, 44], while AdaIN [15] and WCT [26] enabled real-time arbitrary style transfer through feature statistics alignment. However, these frameworks [2, 8] are less generalizable for arbitrary style transfer. Different interesting methods are proposed with the progress of the text-to-image diffusion model. InST [62] inverses a painting into corresponding textual embeddings to guide the text-to-image generative model in creating images of specific artistic appearance. B-LoRA [9] and DEADiff [30] achieve style-content separation by LoRA weight optimization and joint text-image cross-attention layers, respectively. IP-adapter [56] trained a light-weight network to incorporate style embeddings via a cross-attention mechanism. InstantStyle [48] improves the performance by injecting style features into the selective layers of the denoising UNet. CSGO [52] enhances the capacity of the style adapter by training on a curated style dataset. Based on CSGO, StyleStudio [21] proposes the cross-modal AdaIN for harmonizing style and text features. Another method is to perform $K/V$ feature injection [1, 5] or distillation [63] of the reference image at the self-attention layers. Notably, StyleAligned [12] also leverages referenced features as auxiliary information, similar to M3S, it differs in methodology. Their method enforces distributional alignment between target and referenced features through statistical constraints. Such rigid matching can degrade performance and cause content leakage when handling sketches with large structural variance (see Section 4).

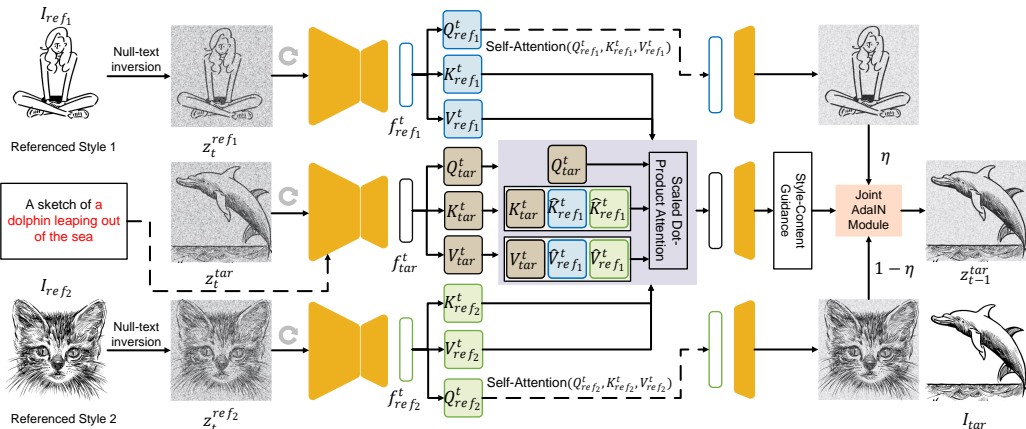

Figure 2: Pipeline of the proposed M3S. Given the referenced style sketches $\mathbf{I}_{ref_1}$ and $\mathbf{I}_{ref_2}$, we invert the two images into the latent space, resulting in latents $\mathbf{z}_t^{ref_1}$ and $\mathbf{z}_t^{ref_2}$. The referenced $K/V$ features are extracted from these latents and employed as auxiliary information in self-attention layers (Section 3.1) for generating target images $\mathbf{I}_{tar}$. A style-content guidance (Section 3.3) is applied to balance the fidelity and style consistency. We apply a joint AdaIN module to control the style tendency (Section 3.2). Generating a single style sketch is a special case in the figure, i.e., blocking out the top or bottom branches.

## 3 Methodology

M3S is a training-free framework for multi-style sketch synthesis that integrates textual prompts with reference style sketches through a pre-trained diffusion backbone. As illustrated in Fig. 2, our method supports single- and multi-style generation by blending style features. The single-style scenario is achieved by disabling one of the branches (lower path in Fig. 2) and setting joint AdaIN coefficients $\eta = 1$. Below, we detail the core components and additional counter-based regular guidance for sparse and abstract freehand sketches.

### 3.1 Style Features Injection

Given referenced sketches $\mathbf{I}_{ref_1}$ and $\mathbf{I}_{ref_2}$, our framework aims to synthesize the target sketch $\mathbf{I}_{tar}$ through controlled feature injection. We first formalize single-style generation by integrating reference features into the denoising steps. At each timestep $t \in (T, ..., 0)$, we extract key-value pairs $(K_{ref_1}, V_{ref_1})$, from the self-attention layer $l$ of the reference sketch's denoising path, while computing target features $(Q_{tar}, K_{tar}, V_{tar})$ in the corresponding layer. The standard attention mechanism operates as:

$$Attention(Q_{tar}, K_{tar}, V_{tar}) = softmax(\frac{Q_{tar}K_{tar}^T}{\sqrt{d_k}})V_{tar}. \tag{1}$$

**Limitations of previous work.** Existing approaches [1, 3, 5] achieve specified-style image generation (or style transfer) through direct feature substitution: $Attention(Q_{ref}, K_{tar}, V_{tar}) \rightarrow Attention(Q_{tar}, K_{ref_1}, V_{ref_1})$. However, this substitution paradigm encounters critical limitations when handling structurally divergent reference-target pairs, particularly for sketches where sparse representations amplify domain gaps. As visualized in column 2 of Fig. 3(a), such methods erroneously focus on local texture replication rather than coherent stroke synthesis, resulting in structural incoherence and artifactual patterns.

**Proposed feature injection approach.** As evidenced in column 4 of Fig. 3(a), our simple yet effective feature concatenation strategy alleviates these limitations. The revised attention computation is mathematically formulated as $Attention\left(Q_{tar}, \begin{bmatrix} K_{tar} \\ K_{ref_1} \end{bmatrix}, \begin{bmatrix} V_{tar} \\ V_{ref_1} \end{bmatrix}\right)$. However, this approach still introduces localized chaotic strokes in certain regions. For natural image synthesis, StyleAligned

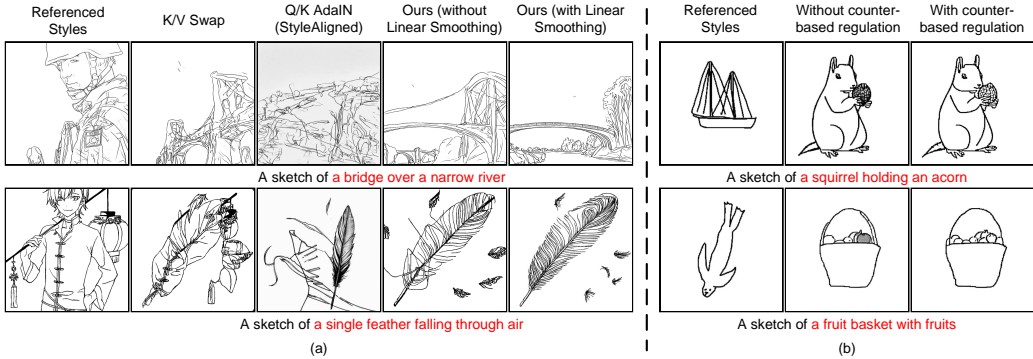

Figure 3: (a) Examples of generated results by different $K/V$ injection method. Direct $K/V$ substitution and AdaIN constraints (i.e., StyleAligned [12]) introduce visual artifacts (chaotic strokes), whereas our feature concatenation strategy improves line quality. Further incorporating linear blending enhances structural coherence by mitigating content leakage. (b) Counter-based regulation guidance (Section 3.4) achieves effective artifact suppression with a controlled trade-off in stroke fidelity.

[12] takes this strategy and attempts to enhance output quality through statistical alignment by modulating: $Q_{tar} = AdaIN(Q_{tar}, Q_{ref_1})$ and $K_{tar} = AdaIN(K_{tar}, K_{ref_1})$, where the AdaIN is

$$AdaIN(x, y) = \sigma(y) \left( \frac{x - \mu(x)}{\sigma(x)} \right) + \mu(y). \tag{2}$$

Unfortunately, different from the scenario of natural images, our experiments (see Fig. 3(a), column 3) demonstrate that such rigid statistical constraints harm sketch generation quality. Instead of this constraint, the key insight underlying M3S is blending deep features from the target content into the reference $K/V$ features, strategically trading style consistency for enhanced visual plausibility. This is realized through a linear smoothing operation with the hyperparameter $\lambda \in [0, 1]$:

$$Attention \left( Q_{tar}, \begin{bmatrix} K_{tar} \\ \hat{K}_{ref_1} \end{bmatrix}, \begin{bmatrix} V_{tar} \\ \hat{V}_{ref_1} \end{bmatrix} \right), \qquad \begin{aligned} \hat{K}_{ref_1} &= \lambda K_{tar} + (1 - \lambda) K_{ref_1}, \\ \hat{V}_{ref_1} &= \lambda V_{tar} + (1 - \lambda) V_{ref_1}. \end{aligned} \tag{3}$$

Increasing $\lambda$ generally enhances aesthetic quality and text alignment. However, as our base model is trained on natural images, excessively high $\lambda$ risks style degradation and naturalistic outputs over sketches. To synthesize sketches combining styles from both $\mathbf{I}_{ref_1}$ and $\mathbf{I}_{ref_2}$, we extend Eq.(3) by simply concatenate additional features $\hat{K}_{ref_2} = \lambda K_{tar} + (1 - \lambda) K_{ref_2}$ and $\hat{V}_{ref_2} = \lambda V_{tar} + (1 - \lambda) V_{ref_2}$. The attention can be written as $Attention(Q_{tar}, [K_{tar}, \hat{K}_{ref_1}, \hat{K}_{ref_2},]^T, [V_{tar}, \hat{V}_{ref_1}, \hat{V}_{ref_2},]^T)$.

## 3.2 Control the Style Tendency

Inspired by [1], we introduce AdaIN modulation for latent noise images to address the color distribution shift problem. This process for single style can be formalized as $\mathbf{z}_t^{tar} = AdaIN(\mathbf{z}_t^{tar}, \mathbf{z}_t^{ref_1})$, where $\mathbf{z}_t^{tar}$ is the latent target image in denoising time step $t$ and $\mathbf{z}_t^{ref_1}$ is obtained by the null-text inversion technology [16]. Next, we discuss the impact of the modulation on sketch style. Assuming a sketch is represented in a bitmap way (0 for strokes, 1 for background), referenced style sketches with dense strokes (e.g., >50% black pixels) exhibit low mean values. AdaIN modulation consequently biases generated sketches toward a lower mean value $\mu$, yielding detailed outputs. Conversely, abstract sketches with sparse strokes demonstrate higher $\mu$, producing more minimalist results. Motivated by this analysis, we introduce a *Joint AdaIN module* for multi-style tendency control:

$$\mathbf{z}_t^{tar} = \eta \cdot AdaIN(\mathbf{z}_t^{tar}, \mathbf{z}_t^{ref_1}) + (1 - \eta) \cdot AdaIN(\mathbf{z}_t^{tar}, \mathbf{z}_t^{ref_2}), \tag{4}$$

where $\eta \in [0, 1]$ is the tendency parameter. Notably, the synthesized results maintain multi-style characteristics even at parameter extremes ($\eta = 0$ or $\eta = 1$), as the self-attention calculation incorporates more than a single stylistic feature.

### 3.3 Style-Content Guidance

While text-to-image diffusion models excel at natural image synthesis, their inherent bias toward photorealistic outputs poses challenges for stylized sketch generation. Conventional classifier-free guidance (CFG) [14] combined with feature injection (Section 3.1) and AdaIN modulation (Section 3.2) struggles to maintain stylistic consistency due to fundamental domain discrepancies between natural images and sketches. To address this, we introduce a null-text conditioned style guidance term that establishes dual control pathways:

$$
\tilde{\epsilon}_t = \epsilon_\theta(\mathbf{z}_t^{tar}, t, \emptyset) + \omega_1 \underbrace{(\epsilon_\theta^\times(\mathbf{z}_t^{tar}, t, text, K_{ref}, V_{ref}) - \epsilon_\theta(\mathbf{z}_t^{tar}, t, \emptyset))}_{content\ guidance\ direction}
$$
$$
+ \omega_2 \underbrace{(\epsilon_\theta^\times(\mathbf{z}_t^{tar}, t, \emptyset, K_{ref}, V_{ref}) - \epsilon_\theta(\mathbf{z}_t^{tar}, t, \emptyset))}_{style\ guidance\ direction},
\tag{5}
$$

where $\epsilon_\theta^\times(\cdot)$ denotes the noise predicted with feature injection (Section 3.1), $\omega_1$ and $\omega_2$ are the content and style guidance scales, respectively. The parameters $\omega_1$ and $\omega_2$ require careful calibration in practice to achieve an optimal balance between style and content. Excessively high values of $\omega_2$ may compromise text-image alignment, while insufficiently low values result in uncontrollable stylistic expression in synthesized sketches.

### 3.4 Counter-based Regulation Guidance

To mitigate potential artifacts in abstract sketch generation with SD v1.5 [33] (Fig. 3(b)), we first apply Tweedie's formula [20] to estimate the denoised latent representation $\mathbf{z}_{0|t}^{tar} = \frac{\mathbf{z}_t^{tar} - \sqrt{1-\bar{\alpha}_t}\tilde{\epsilon}_t}{\sqrt{\bar{\alpha}_t}}$ [13, 40], which is decoded to image $\mathbf{I}_{tar}^{0|t}$. Subsequently, we extract directional gradients through Sobel operators [39]: $grad_x = S_x * \mathbf{I}_{tar}^{0|t}, grad_y = S_y * \mathbf{I}_{tar}^{0|t}$, where $*$ denotes convolution. Then, we calculate the regular term loss and optimize the latent representations as

$$
\mathcal{L}_{edge} = -|grad_x| - |grad_y|, \quad \mathbf{z}_{0|t}^{tar} = \mathbf{z}_{0|t}^{tar} - \gamma \nabla \frac{\mathcal{L}_{edge}}{\mathbf{z}_{0|t}^{tar}}.
\tag{6}
$$

The updated $\mathbf{z}_{0|t}^{tar}$ is subsequently applied to calculate $\mathbf{z}_{t-1}^{tar}$ with DDIM step [40]. In practice, we empirically set $\gamma = 60$ as the default and clamp $grad_x$ and $grad_y$ within $[-0.001, 0.001]$ to prevent over-amplification of gradient magnitudes that could degrade generation quality.

## 4 Experiments

**Dataset and Metrics**   We evaluate different methods for single-style referenced generation on six diverse sketch datasets encompassing professional, amateur, and abstract styles: four professional styles from 4skst [37] (Styles 1-4, artist-drawn with referenced images), a web-collected diverse style set (Style 5, 20 sketches from open-source platforms), and 50 abstract freehand sketches from Sketchy [35] (Style 6). For systematic testing, we generate 50 textual prompts via DeepSeek [27] using the template "A sketch of ...", pairing each prompt with a randomly selected referenced sketch. To evaluate the performance of M3S and baselines, CLIP-T [32] is used to measure the alignment between generated sketches and prompts. For style consistency, we use the similarity between referenced and target images with the extracted features from DINO [61]. Similar to [1], the distance of Gram matrices calculated from VGG [38] is also considered.

**Implementation Details**   We implement M3S on both Stable Diffusion v1.5 [33] and SDXL [29], with all visual results in this paper generated by M3S (SD v1.5) unless otherwise specified (more M3S (SDXL)'s results are in the Appendix). For M3S (SD v1.5), we configure $\omega_1 = 15$, $\omega_2 = 15$ and $\lambda = 0.1$ Styles 1-5, while using $\omega_1 = 15$, $\omega_2 = 25$ and $\lambda = 0.05$ for Style 6. Similar to [1], feature injection is applied to self-attention layers in the UNet decoder where feature map resolutions are $32 \times 32$ and $64 \times 64$. For M3S (SDXL), we set $\omega_1 = 15$, $\omega_2 = 15$ and $\lambda = 0.1$ for Styles 1-5, adjusting to $\omega_1 = 7.5$, $\omega_2 = 20$ and $\lambda = 0.05$ for Style 6. The specific injection layers are detailed in the Appendix. The $\omega_2$ parameter linearly increases from $\omega_2/3$ to $\omega_2$ throughout the denoising process in both implementations. We applied DDIM [40] to sample the target sketches with 100 steps.

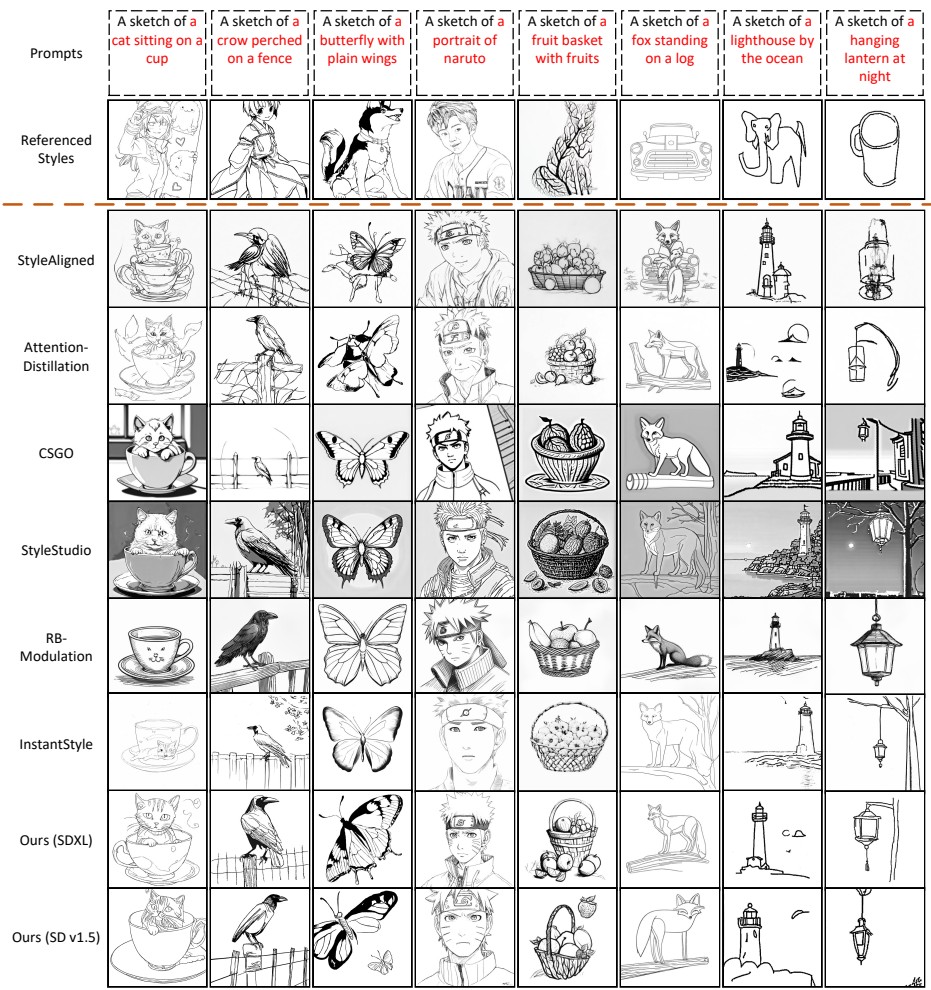

Figure 4: Qualitative comparison of different methods. Most evaluation cases are challenging cross-domain synthesis scenarios. The referenced images in columns 1-4 are from Style 1-4, columns 5-6 are from Style 5, and columns 7-8 are from Style 6.

**Baselines** We compare with state-of-the-art (SOTA) methods spanning diverse technical paradigms: StyleAligned [12] (self-attention feature injection), RB-Modulation [34] (multi-attention aggregation), AttentionDisillation [63], and methods leveraging specialized style adapters — InstantStyle [48], CSGO [52], and StyleStudio [21]. These methods are primarily designed for single-reference scenarios. All baselines are implemented with their open-source codes.

## 4.1 Qualitative Analysis

In Fig. 4, we illustrated the qualitative synthesized results in various styles with different methods. Our method effectively captures referenced style sketch attributes, including stroke thickness, pixel density, and luminance levels, to achieve zero-shot text-aligned sketch synthesis. Compared to StyleAligned [12] and AttentionDistillation [63], which preserve stylistic fidelity at the cost of content leakage (e.g., the fox erroneously positioned on a car in the third to last column) and chaotic stroke patterns (visible in AttentionDistillation's first three columns), our M3S framework maintains strict content-semantic consistency while eliminating structural artifacts. These results demonstrate that excessive feature constraints may degrade sketch quality, particularly in artist-style sketches. Although methods like InstantStyle [48] and CSGO [52] generate aesthetically plausible results aligned with textual prompts, they exhibit discernible deviations from referenced styles and constrained style diversity compared to our approach. This phenomenon likely stems from these

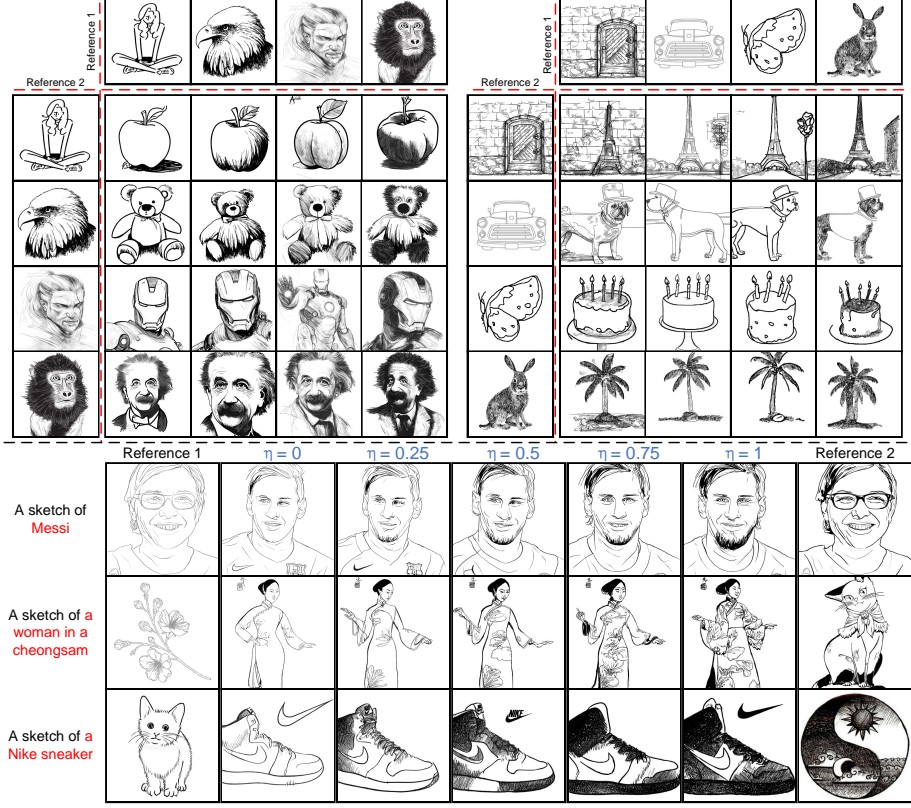

Figure 5: Examples of generated sketches with two referenced style images. **Top**: Same prompts are used in each row, and the prompts are in the Appendix. We set the style tendency $\eta = 0.5$ in these cases. **Bottom**: Results of different $\eta$ to control the style tendency.

methods' primary focus on natural image style transfer, where their feature injection mechanisms predominantly operate in text space rather than visual space.

The multi-style fusion capability is partially illustrated through specific examples in Fig. 5, and there are some interesting cases: (1) The apple sketch (top-left) fuses distinct stylistic attributes, clear black contours from one reference, and localized texturing patterns from another. (2) The birthday cake synthesis (top-right). When fusing an abstract freehand butterfly sketch and a precisely structured car drawing, the result inherits the butterfly's stroke coloration and the car's orderly linework. Conversely, the cake's contours exhibit amateurish characteristics when substituting the car reference with other sketches. These examples illustrate our framework's capacity to selectively blend stylistic elements across references while preserving domain-appropriate stroke characteristics. Our method also enables flexible style interpolation through the Joint AdaIN module, as shown at the bottom of Fig. 5. These interpolated results may be able to stimulate the user's creativity. For instance, increasing $\eta$ progressively enhances the definition of Messi's beard strokes.

## 4.2 Quantitative Analysis

Table 1 reports the quantitative results of different methods. Under default parameter settings, our M3S (SDXL) achieves the best average CLIP score of 0.3514, demonstrating superior text alignment. While the DINO score and VGG style loss slightly trail AttentionDistillation [63], this reflects our method's balanced style-content trade-off. To validate flexibility, we provide additional results with style-oriented parameters (denoted as Ours (SDXL*) in the table), where reducing content-guidance $\omega_1$ and increasing style-guidance $\omega_2$ yields style metrics comparable to AttentionDistillation while slightly retaining CLIP score advantages. This highlights M3S's user-adjustable controllability for subjective style-content balancing. Notably, methods like StyleStudio [21] produce high-quality images but exhibit lower CLIP scores, likely due to their outputs resembling natural images over

Table 1: Sketch-text alignment and style consistency performance comparison across styles. 'Ours (SDXL$^*$)' denotes that the parameters of our method are set to $\omega_1 = 7.5, \omega_2 = 20$, and $\lambda = 0.0$.

| Method | Style1 | | | Style2 | | | Style3 | | |
|---|---|---|---|---|---|---|---|---|---|
| | CLIP-T($\uparrow$) | DINO($\uparrow$) | VGG($\downarrow$) | CLIP-T($\uparrow$) | DINO($\uparrow$) | VGG($\downarrow$) | CLIP-T($\uparrow$) | DINO($\uparrow$) | VGG($\downarrow$) |
| StyleAligned [12] | 0.3130 | 0.6691 | 0.0308 | 0.3095 | 0.7064 | 0.0684 | 0.3013 | 0.6309 | 0.0621 |
| AttentionDistillation [63] | 0.3305 | **0.7738** | **0.0930** | 0.3320 | **0.7724** | 0.0320 | 0.3225 | **0.7132** | **0.0305** |
| CSGO [52] | 0.3336 | 0.5276 | 0.0571 | 0.3257 | 0.5409 | 0.1370 | 0.3232 | 0.5154 | 0.1018 |
| StyleStudio [21] | 0.3395 | 0.5164 | 0.1873 | 0.3351 | 0.5601 | 0.1954 | 0.3349 | 0.5337 | 0.1790 |
| RB-Modulation [34] | 0.3298 | 0.3624 | 0.0592 | 0.3300 | 0.3429 | 0.2085 | 0.3279 | 0.3453 | 0.1733 |
| InstantStyle [48] | 0.3512 | 0.4934 | 0.0417 | 0.3508 | 0.4929 | 0.1577 | **0.3455** | 0.4394 | 0.1321 |
| Ours (SDXL) | **0.3607** | 0.6545 | 0.0165 | **0.3556** | 0.6531 | 0.0674 | 0.3422 | 0.6041 | 0.0534 |
| Ours (SD v1.5) | 0.3507 | 0.6383 | 0.0200 | 0.3452 | 0.6846 | 0.0616 | 0.3416 | 0.6269 | 0.0571 |
| Ours (SDXL*) | 0.3480 | 0.7344 | 0.0122 | 0.3340 | 0.7356 | 0.0464 | 0.3319 | 0.6870 | 0.0371 |

| | Style4 | | | Style5 | | | Style6 | | |
|---|---|---|---|---|---|---|---|---|---|
| | CLIP-T($\uparrow$) | DINO($\uparrow$) | VGG($\downarrow$) | CLIP-T($\uparrow$) | DINO($\uparrow$) | VGG($\downarrow$) | CLIP-T($\uparrow$) | DINO($\uparrow$) | VGG($\downarrow$) |
| StyleAligned [12] | 0.3137 | 0.6407 | 0.0244 | 0.3004 | 0.5428 | 0.0445 | 0.2879 | 0.4445 | 0.0300 |
| AttentionDistillation [63] | 0.3222 | **0.7572** | **0.0061** | 0.3377 | **0.6221** | **0.0173** | 0.3289 | 0.7027 | 0.0190 |
| CSGO [52] | 0.3321 | 0.5134 | 0.0526 | 0.3298 | 0.4288 | 0.0972 | 0.3241 | 0.5012 | 0.0716 |
| StyleStudio [21] | 0.3402 | 0.5100 | 0.1595 | 0.3377 | 0.3539 | 0.1215 | 0.3338 | 0.3612 | 0.1434 |
| RB-Modulation [34] | 0.3178 | 0.3373 | 0.0465 | 0.3247 | 0.3233 | 0.0972 | 0.3221 | 0.2737 | 0.0780 |
| InstantStyle [48] | 0.3513 | 0.4494 | 0.0262 | 0.3480 | 0.4408 | 0.0601 | 0.3417 | 0.5130 | 0.0421 |
| Ours (SDXL) | **0.3612** | 0.6493 | 0.0115 | 0.3467 | 0.5332 | 0.0304 | **0.3420** | 0.6922 | 0.0259 |
| Ours (SD v1.5) | 0.3518 | 0.6337 | 0.0136 | **0.3494** | 0.5777 | 0.0272 | 0.3405 | **0.7653** | **0.0170** |
| Ours (SDXL*) | 0.3506 | 0.7212 | 0.0085 | 0.3383 | 0.6328 | 0.0191 | - | - | - |

Table 2: The average rating of different methods by the human preference assessment.

| | StyleAligned [12] | AttentionDistillation [63] | CSGO [52] | StyleStudio [21] |
|---|---|---|---|---|
| Rating | 2.77 | 4.28 | 3.83 | 4.22 |
| | RB-Modulation [34] | InstantStyle [48] | Ours(SD v1.5) | Ours (SDXL) |
| Rating | 4.20 | 5.08 | 5.44 | **6.19** |

sketches. Similarly, lower CLIP scores in Style 3 may stem from dense black patches in references, whereas Style 4 shows the opposite trend. Each sketch takes about 40 seconds (M3S (SD v1.5)) and 70 seconds (M3S (SDXL)) on an A100 40GB GPU.

We conducted **a human preference assessment** via structured questionnaires to evaluate text-to-sketch synthesis performance. Each questionnaire contained: 1) Six randomly selected sets of generated sketches. 2) Eight anonymized outputs per set from different models under identical prompts and reference styles. 3) Evaluation criteria: Comprehensive assessment across three dimensions—text alignment, style consistency, and generation quality. Participants ranked results on a scale of 1-8 (8=optimal). From 58 submissions (avg. completion: 4m16s), we excluded 14 invalid responses (<60s completion/missing rankings), retaining 44 validated questionnaires. The results are shown on Table 2. M3S (SDXL) achieved the highest average score (6.19), and M3S (SD v1.5) secured a strong performance (5.44). We evaluate the statistical significance by rank tests, revealing that M3S (SD v1.5) significantly outperforms all baseline methods except InstantStyle (p-value=0.26). When we align the backbone with InstantStyle (i.e., SDXL), our M3S (SDXL) demonstrates superiority over it (p-value=$1.06 \times 10^{-5}$).

The **quantitative results of multi-style generation** are shown in Table 3. We generate outputs for each prompt by randomly selecting two reference sketches from the Style 5 (S5) dataset. To specifically validate generation performance with significantly distinct reference styles, we conducted an additional experiment set pairing one randomly selected S5 image with one randomly chosen image from the QuickDraw (QD) dataset [11] per prompt. When the references exclusively originate from S5, M3S maintains text alignment comparable to single-style generation. For style consistency, DINO-ref1 exhibits a positive correlation, while DINO-ref2 shows a negative correlation. At boundary conditions ($\eta = 0$ or 1) of multi-style sketch generation, only one style participates in AdaIN modulation for image generation. For the two reference styles, replacing the pair of S5-S5 with QD-S5 reduces style consistency for both implementations (i.e., SD v1.5 and SDXL), though M3S (SD v1.5) demonstrates superior robustness. Crucially, M3S (SDXL) struggles to effectively utilize QD's abstract features, evidenced by significantly lower scores of DINO-ref1 than DINO-ref2 in QD-S5 pairs. This limitation stems from SDXL's high-fidelity optimization [29] - its user-tested superiority over SD v1.5 creates inherent incompatibility with low-quality, abstract datasets like QD.

Table 3: Multi-style sketch generation performance under different $\eta$ values.

| M3S Imp. | Ref. style | $\eta = 0$ | | | $\eta = 0.25$ | | | $\eta = 0.5$ | | |
|---|---|---|---|---|---|---|---|---|---|---|
| | | CLIP-T($\uparrow$) | DINO-ref1($\uparrow$) | DINO-ref2($\uparrow$) | CLIP-T($\uparrow$) | DINO-ref1($\uparrow$) | DINO-ref2($\uparrow$) | CLIP-T($\uparrow$) | DINO-ref1($\uparrow$) | DINO-ref2($\uparrow$) |
| SDXL | S5-S5 | 0.3442 | 0.3936 | 0.4944 | 0.3514 | 0.4180 | 0.4821 | 0.3495 | 0.4408 | 0.4556 |
| SD v1.5 | S5-S5 | 0.3465 | 0.3850 | 0.4776 | 0.3453 | 0.4215 | 0.4597 | 0.3499 | 0.4469 | 0.4509 |
| SDXL | QD-S5 | 0.3426 | 0.3051 | 0.4724 | 0.3455 | 0.3266 | 0.4622 | 0.3457 | 0.3330 | 0.4397 |
| SD v1.5 | QD-S5 | 0.3434 | 0.3630 | 0.4339 | 0.3417 | 0.3948 | 0.4236 | 0.3452 | 0.4102 | 0.4057 |
| | | $\eta = 0.75$ | | | $\eta = 1$ | | | | | |
| | | CLIP-T($\uparrow$) | DINO-ref1($\uparrow$) | DINO-ref2($\uparrow$) | CLIP-T($\uparrow$) | DINO-ref1($\uparrow$) | DINO-ref2($\uparrow$) | | | |
| SDXL | S5-S5 | 0.3499 | 0.4578 | 0.4221 | 0.3470 | 0.4693 | 0.3975 | | | |
| SD v1.5 | S5-S5 | 0.3478 | 0.4528 | 0.4257 | 0.3528 | 0.4626 | 0.3825 | | | |
| SDXL | QD-S5 | 0.3447 | 0.3409 | 0.4209 | 0.3396 | 0.3617 | 0.3916 | | | |
| SD v1.5 | QD-S5 | 0.3440 | 0.4250 | 0.3938 | 0.3468 | 0.4381 | 0.3766 | | | |

Table 4: Results of ablation experiments with different levels of feature injection.

| | K/V Swap | $\lambda = 0$ | $\lambda = 0.05$ | $\lambda = 0.1$ | $\lambda = 0.15$ | $\lambda = 0.2$ | $\lambda = 0.25$ | $\lambda = 1$ |
|---|---|---|---|---|---|---|---|---|
| CLIP-T($\uparrow$) | 0.3120 | 0.3409 | 0.3443 | **0.3473** | 0.3457 | 0.3468 | 0.3458 | 0.3382 |
| DINO($\uparrow$) | **0.8109** | 0.7437 | 0.6978 | 0.6459 | 0.6161 | 0.5904 | 0.5719 | 0.3707 |
| aesthetic | 4.5750 | 4.7952 | 4.8892 | 4.9488 | 4.9577 | 4.9843 | 5.0057 | 5.0493 |

## 4.3 Ablation Study

We conduct the ablation study on styles 1-4 with SD V1.5 as the backbone. To evaluate the effectiveness of our feature injection approach, Table 4 reports the average scores across these styles. Aesthetic metric [36] is included to reflect the effect more comprehensively. When employing basic K/V Swap, the output maintains excellent style consistency with a DINO score of 0.8109, but its low CLIP score reflects poor alignment between generated sketches and textual prompts.

By incorporating reference features as auxiliary inputs (i.e., $\lambda = 0$), the CLIP metric improves by 9.26%, accompanied by an acceptable reduction in DINO. However, the aesthetic score of generated images under this configuration drops to 4.7952, significantly lower than the reference style average of 5.0549. Increasing $\lambda$ mitigates aesthetic degradation and improves text alignment, but further erodes style fidelity. Beyond a critical $\lambda$, CLIP gains diminish while style loss intensifies, leading to our experimentally determined default of $\lambda = 0.1$ as an optimal compromise. Fig. 6 demonstrates the impact of content guidance ($\omega_1$) and style guidance ($\omega_2$) on generation results. When $\omega_1$ is fixed, increasing $\omega_2$ reduces the number of intersections in curvilinear textures on the deer's body,

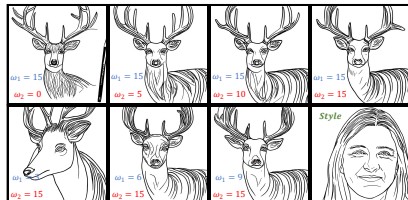

Figure 6: The generated results with different content control scale $\omega_1$ and style control scale $\omega_2$.

driving the output closer to the reference style. Conversely, enhancing $\omega_1$ under a fixed $\omega_2$ improves the visual quality of the generated deer. More ablation results are in the Appendix.

## 5 Conclusions

We propose a novel training-free framework named M3S for zero-shot sketch synthesis blending different styles. The key insight in our method is to take the referenced features as auxiliary information and modulate the latent noise images with the joint AdaIN module. This modulated approach enables the user to control the style tendency. Additionally, the style-content direction guidance provides the flexibility to balance the fidelity and style consistency. A direction for future research and the functionality not included for now is achieving localized style control, where users can explicitly assign specific styles to particular regions rather than relying on the model's automatic style assignment. This capability would further assist and inspire artistic creation.

## Acknowledgement

This work was supported by Shanghai Municipal Science and Technology Major Project, China (Grant No. 2021SHZDZX0102).

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

# Appendix

The appendix is organized into several sections, including more analysis and additional details. These topics are as follows:

- In Section A, we provide preliminaries about diffusion model.
- In Section B, more implementation details of our M3S are provided.
- In Section C, we analyze the effectiveness of the regular term for abstract and sparse sketch generation.
- In Section D, we present the influence of the content guidance $\omega_1$ and the style guidance $\omega_2$.
- In Section E, we discuss the effectiveness of linear smoothing for style injection.
- In Section F, we compare other generative model for sketch synthesis.
- In Section G, we provide more synthesized results of M3S (SDXL).
- In Section H, we discuss the limitations of M3s.
- In Section I, we discuss the potential social impact of our method.
- In Section J, we show the selected images of Style 5.
- In Section K, we provide the prompts used for qualitative analysis.
- In Section L, we present the prompts for Fig. 5's multi-style generation.

## A   Preliminaries

### A.1   Denoising Diffusion Probabilistic Models (DDPM)

Denoising Diffusion Probabilistic Models (DDPMs) [13] establish a generative framework that learns to model the data distribution $q_{data}(\boldsymbol{x}_0)$ through two complementary phases:

**Forward Diffusion Process.**   The forward process systematically perturbs data samples through progressive noise injection. Given initial data $\boldsymbol{x}_0 \sim q_{data}(\boldsymbol{x}_0)$, it constructs a Markov chain of latent variables $\{\boldsymbol{x}_t\}_{t=1}^{T}$ by gradually adding Gaussian noise according to a predefined schedule $\{\beta_t\}_{t=1}^{T} \in (0,1)$:

$$q(\boldsymbol{x}_{1:T}|\boldsymbol{x}_0) = \prod_{t=1}^{T} \mathcal{N}\left(\boldsymbol{x}_t; \sqrt{1-\beta_t}\boldsymbol{x}_{t-1}, \beta_t \boldsymbol{I}\right) \tag{7}$$

This transforms the complex data distribution into an isotropic Gaussian distribution $q(\boldsymbol{x}_T) \approx \mathcal{N}(\boldsymbol{0}, \boldsymbol{I})$.

**Reverse Denoising Process.**   The reverse process learns to invert the diffusion trajectory by iteratively denoising from $\boldsymbol{x}_T \sim \mathcal{N}(\boldsymbol{0}, \boldsymbol{I})$. Since the true reverse transition $q(\boldsymbol{x}_{t-1}|\boldsymbol{x}_t)$ depends on the intractable data distribution $q_{data}(\boldsymbol{x}_0)$, DDPMs approximate it through a learned conditional Gaussian:

$$p_\theta(\boldsymbol{x}_{t-1}|\boldsymbol{x}_t) = \mathcal{N}\left(\boldsymbol{x}_{t-1}; \boldsymbol{\mu}_\theta(\boldsymbol{x}_t, t), \boldsymbol{\Sigma}_\theta(\boldsymbol{x}_t, t)\right) \tag{8}$$

**Training Objective.**   Instead of direct mean prediction, DDPMs adopt a noise prediction parameterization. For timestep $t$ uniformly sampled from $\{1, ..., T\}$, the network $\boldsymbol{\epsilon}_\theta$ predicts the injected noise through:

$$\mathcal{L}_{simple} = \mathbb{E}_{\boldsymbol{x}_0, \epsilon, t}\|\boldsymbol{\epsilon} - \boldsymbol{\epsilon}_\theta(\sqrt{\bar{\alpha}_t}\boldsymbol{x}_0 + \sqrt{1-\bar{\alpha}_t}\boldsymbol{\epsilon}, t)\|^2 \tag{9}$$

where $\alpha_t := 1 - \beta_t$ and $\bar{\alpha}_t := \prod_{s=1}^{t} \alpha_s$. The denoised mean is then derived as:

$$\boldsymbol{\mu}_\theta(\boldsymbol{x}_t, t) = \frac{1}{\sqrt{\alpha_t}}\left(\boldsymbol{x}_t - \frac{\beta_t}{\sqrt{1-\bar{\alpha}_t}}\boldsymbol{\epsilon}_\theta(\boldsymbol{x}_t, t)\right) \tag{10}$$

During inference, DDPMs sample from $p_\theta(\boldsymbol{x}_{t-1}|\boldsymbol{x}_t)$ iteratively from $t = T$ to $t = 1$. The complete derivation and connections to stochastic differential equations are detailed in [42].

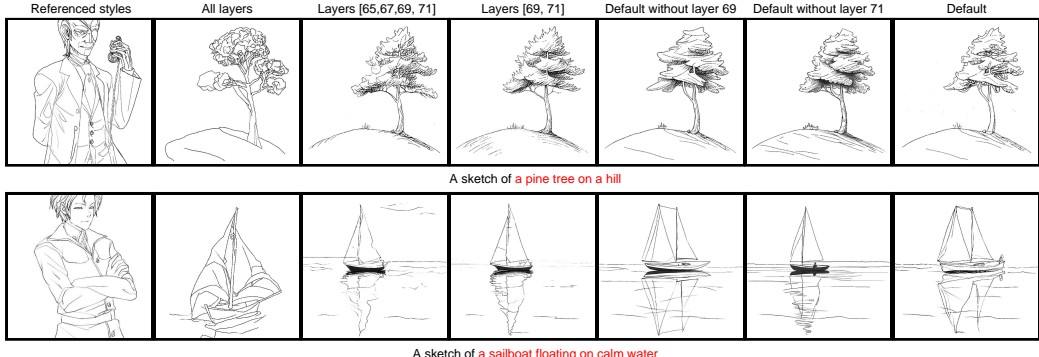

| Referenced styles | All layers | Layers [65,67,69, 71] | Layers [69, 71] | Default without layer 69 | Default without layer 71 | Default |

A sketch of a pine tree on a hill

A sketch of a sailboat floating on calm water

Figure 7: Examples of generated sketches with different settings of feature-injection layers.

## B    More Implementation Details

To eliminate the possible localized shadows in the generated sketches, we followed the practice of [55] and performed a brightening operation on the images. Specifically, we set pixels with values exceeding 0.7 to 1, where the pixel value range is normalized between -1 and 1.

Our M3S (SDXL) selects some self-attention layers from the upsample UNet branch to perform the feature injection. We first enumerate all attention layers in the decoder, where odd-numbered indices correspond to self-attention layers. In practice, layers with indices [1,9,17,25,33,41,49,57,69,71] are selected for reference style information injection. The examples in Fig. 7 demonstrate the critical role of the final self-attention layer in style control. When feature injection is disabled at layer 71, the generated images overemphasize textual prompts, resulting in excessive color patches that deviate from the reference style. Conversely, injecting features across all self-attention layers achieves exceptional style consistency but compromises text alignment (e.g., excessive folds in generated sailboat sails). This indicates our layer selection strategy inherently balances style-content trade-offs. Users can adopt the default configuration or customize layer choices based on visual examples to meet specific needs.

## C    Analysis of the Regular Term

The regularization term in Eq. (6) primarily serves to suppress shadows and enhance stroke definition. As demonstrated in Fig. 8, increasing hyperparameter $\gamma$ progressively diminishes shadow regions while accentuating strokes, albeit with unintended over-sharpening that introduces stroke unevenness. Empirical analysis reveals optimal performance when $\gamma$ resides within [40, 60], balancing shadow removal and stroke smoothness. Applying this technique to enhance abstract sketches introduces approximately 5 seconds of additional processing time per image.

## D    Analysis of the Style-Content guidance

Fig. 9 demonstrates the effects of content guidance ($\omega_1$) and style guidance ($\omega_2$) through professional and abstract style examples. For the professional style (left): (1) All configurations yield text-compliant sketches. (2) High $\omega_1$ values (20/25) introduce extraneous pens, likely due to insufficient style guidance during early denoising (note $\omega_2$'s linear scheduling in Paragraph Implementation Details). (3) Consistent with Fig. 6, we observe that higher $\omega_1$ enhances aesthetics (e.g., streamlined deer faces) and increased $\omega_2$ improves style adherence (rule-based non-intersecting fur strokes). (4) Balancing trade-offs: At $\omega_1 = 5$, elevating $\omega_2$ causes structural distortion (deer torso deformation at $\omega_2 = 20$). This is mitigated by increasing $\omega_1$, motivating our default $\omega_1 = 15, \omega_2 = 15$.

For abstract styles (right): (1) $\omega_1 = 5, \omega_2 = 5$ produces over-detailed faces inconsistent with the referenced style sketch. (2) Higher $\omega_2$ eliminates excessive details but weakens "yoga" characteristics. We therefore set $\omega_1 = 15, \omega_2 = 25$ for abstract cases, accepting marginal quality degradation to prevent artifacts like striped pants (visible at $\omega_1 = 15, \omega_2 = 15$).

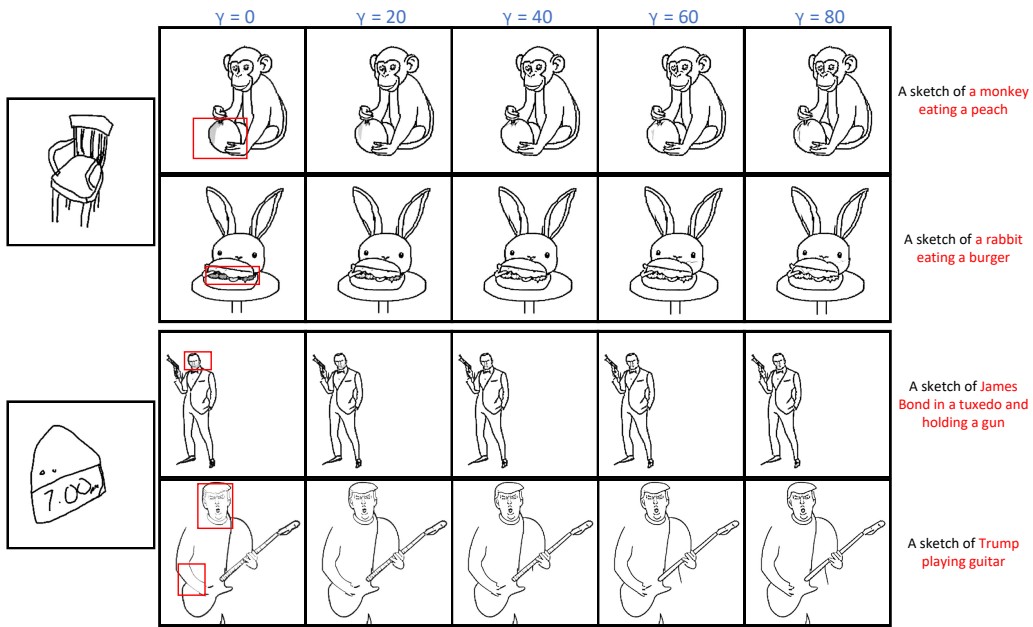

Figure 8: The effectiveness of different $\gamma$ of eq. (6). Areas demarcated in red signify regions of substantial shading or lines that are less clearly defined.

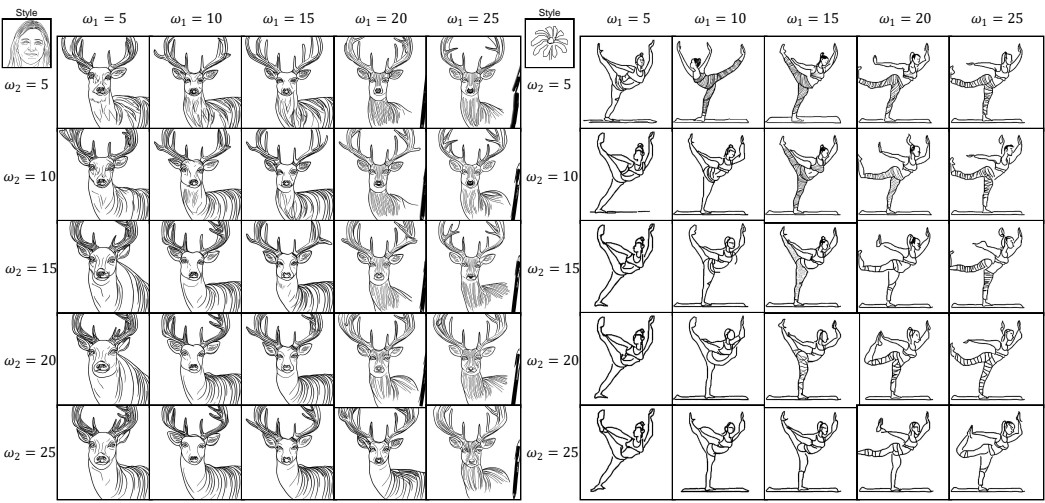

Figure 9: The generated results with different content guidance scale $\omega_1$ and style guidance scale $\omega_2$. Left: "a sketch of a deer". Right: "a sketch of a person doing yoga".

## E   Analysis of Linear Smoothing for Feature Injection

To further analyze the impact of linear smoothing parameter $\lambda$, Fig. 10 visualizes generation results under varying $\lambda$ values, while Table 5 extends the quantitative analysis from Table 4 with complete metrics. Key observations include: (1) $\lambda = 0$: Excessive focus on reference style introduces chaotic strokes that compromise text alignment (e.g., window artifacts in row 2). (2) $\lambda \in (0, 0.25)$: Progressive artifact reduction improves visual cleanliness (row 3 fox torso refinement) while retaining core style attributes. (3) $\lambda = 0.25$: Incipient style degradation manifests as faint roof lines contradicting reference patterns (row 2). (4) $\lambda = 1$: Complete style disengagement yields naturalistic images lacking artistic stylization. Table 5 quantitatively corroborates these findings: decreasing DINO

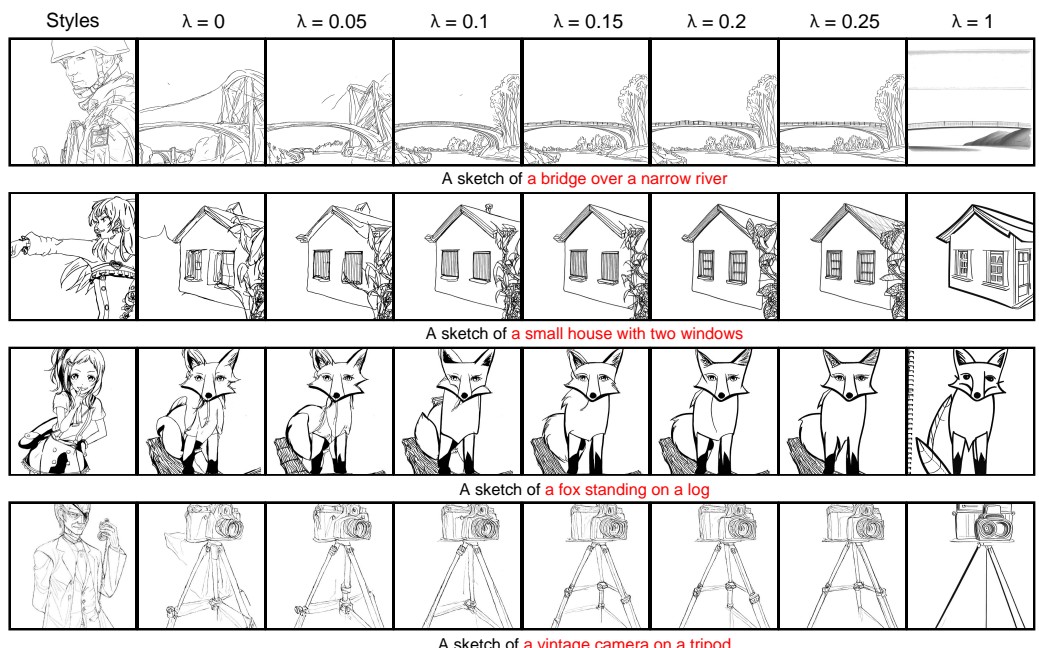

| Styles | λ = 0 | λ = 0.05 | λ = 0.1 | λ = 0.15 | λ = 0.2 | λ = 0.25 | λ = 1 |

A sketch of a bridge over a narrow river

A sketch of a small house with two windows

A sketch of a fox standing on a log

A sketch of a vintage camera on a tripod

Figure 10: Examples of different linear smoothing parameter $\lambda$ for feature injection.

Table 5: Results of ablation experiments with different levels of feature injection. This Table reports the source data of Table 4.

| Method | Style1 | | | Style2 | | | Style3 | | | Style4 | | |
|---|---|---|---|---|---|---|---|---|---|---|---|---|
| | CLIP-T(↑) | DINO(↑) | aesthetic | CLIP-T(↑) | DINO(↑) | aesthetic | CLIP-T(↑) | DINO(↑) | aesthetic | CLIP-T(↑) | DINO(↑) | aesthetic |
| K/V swap | 0.3103 | **0.8132** | 4.5519 | 0.3123 | **0.8474** | 4.4587 | 0.3029 | **0.7854** | 4.6793 | 0.3225 | **0.7977** | 4.6101 |
| λ = 0 | 0.3437 | 0.7452 | 4.7498 | 0.3390 | 0.7707 | 4.7506 | 0.3343 | 0.7315 | 4.8840 | 0.3467 | 0.7273 | 4.7962 |
| λ = 0.05 | 0.3474 | 0.6876 | 4.8609 | 0.3449 | 0.7297 | 4.8636 | 0.3360 | 0.6816 | 4.9494 | 0.3487 | 0.6923 | 4.8829 |
| λ = 0.1 | **0.3507** | 0.6383 | 4.9555 | 0.3452 | 0.6846 | 4.9043 | **0.3416** | 0.6269 | 4.9877 | **0.3518** | 0.6337 | 4.9476 |
| λ = 0.15 | 0.3486 | 0.6153 | 4.9473 | **0.3463** | 0.6396 | 4.9113 | 0.3382 | 0.6005 | 5.0179 | 0.3497 | 0.6090 | 4.9544 |
| λ = 0.2 | 0.3501 | 0.5971 | 5.0359 | 0.3463 | 0.6170 | 4.9240 | 0.3404 | 0.5735 | 4.9915 | 0.3503 | 0.5741 | 4.9856 |
| λ = 0.25 | 0.3482 | 0.5746 | 5.0194 | 0.3443 | 0.6015 | 4.9610 | 0.3400 | 0.5516 | 5.0153 | 0.3506 | 0.5597 | 5.0269 |
| λ = 1 | 0.3398 | 0.3702 | 5.0199 | 0.3382 | 0.3694 | 5.0608 | 0.3352 | 0.3870 | 5.0736 | 0.3396 | 0.3563 | 5.0428 |

scores and improving aesthetic metrics with increasing $\lambda$, reflecting the inherent trade-off between style preservation and content alignment.

## F    Comparisons with More Methods

While the main text focuses on style-specific generation methods, Fig. 11 provides a comparative analysis with broader sketch synthesis approaches. Implementations of CLIPasso [46] and DiffSketcher [53] follow their official codebases, requiring over 3 minutes per sketch generation. We observe that CLIPasso, DiffSketcher, and SketchRNN [11] produce stylistically homogeneous outputs. Notably, CLIPasso's reliance on foreground extraction networks leads to chaotic strokes when segmentation fails (e.g., indistinguishable floral patterns). Although Stable Diffusion v1.5 [33] exhibits style diversity, its outputs demonstrate uncontrolled style variation and struggle to produce sketches with white backgrounds and black strokes. Our method effectively addresses these limitations through controlled style injection while maintaining superior visual quality. When using DiffSketcher's outputs as reference styles, M3S successfully emulates this style (Fig.11-bottom), demonstrating seamless integration with specialized models.

## G    Generated Sketches by M3S (SDXL)

Compared to SD v1.5, SDXL employs a significantly larger denoising network architecture and enhances the dimensionality of text-conditioning embeddings. In the main text, we provide several

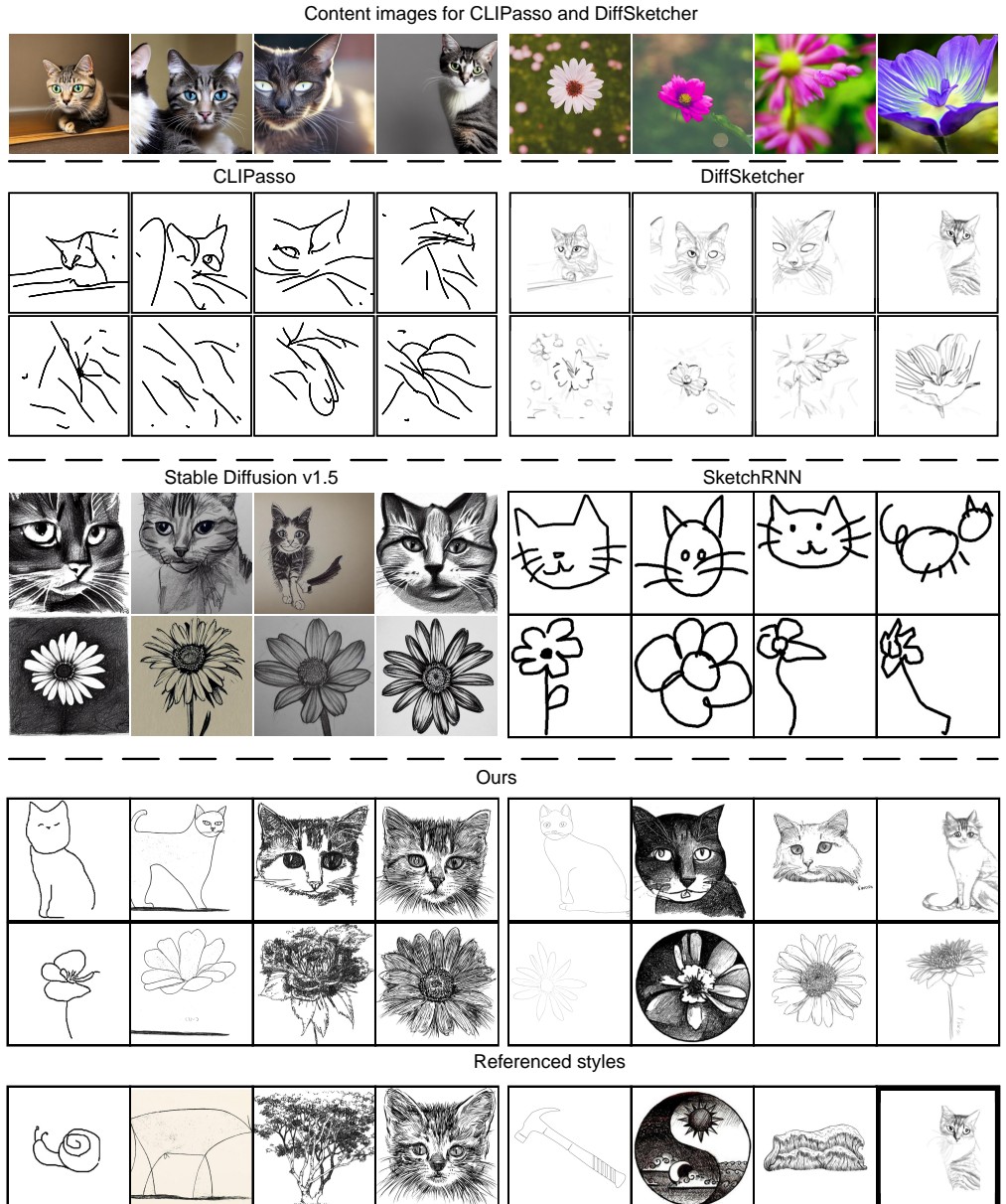

Figure 11: Examples with different methods to generate cats and flowers. CLIPasso [46] and DiffSketcher [53] require the content image to extract sketches. The prompt 'a sketch of a cat/flower' is used for Stable Diffusion V1.5 [33] and our method. In the last column of our method, we use one of the synthesized sketches of DiffSketcher as the referenced style.

results synthesized by M3S (SD v1.5), and we illustrate generated sketches by the more powerful M3S (SDXL).

Benefiting from SDXL's superior text-to-image synthesis capabilities, our M3S framework achieves precise text-aligned sketch generation across diverse artistic styles, even for complex compositional prompts like `"A sketch of a cyberpunk-style cat with mechanical limbs"`, while maintaining strict adherence to reference style characteristics, as shown in Fig. 12. Fig. 13 illustrates the results of multi-style generation.

M3S (SDXL) effectively integrates styles from dual reference images through its joint AdaIN modulation mechanism for style tendency control. The mirror downside is that the M3S (SDXL) is less than

perfect in the style migration of local details. For instance, the elephant contour in row 2 demonstrates insufficient adoption of the left reference's segmented-line style, where discrete strokes appear less pronounced. This discrepancy likely stems from SDXL's architectural advancements: deeper network layers, augmented attention modules, and heightened model complexity. Notably, these modifications simultaneously enhance text-alignment precision, particularly for prompts requiring fine-grained semantic grounding.

M3S (SDXL) preserves the intrinsic diversity of diffusion models — generating distinct sketches from identical textual prompts and reference styles through stochastic initial noise sampling, as demonstrated in Fig. 14. This stochasticity manifests across multiple dimensions: (1) Form variation. Divergent animal pose. (2) Facial Articulation. Unique combinations of facial features and hairstyles. (3) Compositional novelty. Alternative spatial arrangements of scene elements. Such multi-faceted diversity provides artists with rich creative inspiration to create sketches.

## H   Limitation

Fig. 15 Although our M3S demonstrates superior performance in most scenarios, certain limitations persist. As shown in the left panel of Fig. 15, when reference sketches are excessively sparse with content concentrated in small image regions, M3S struggles to generate complete and clear sketches. This stems from two factors: (1) Extreme sparsity hinders effective style feature extraction for guiding generation, and (2) Higher pixel intensity averages in such references disrupt AdaIN modulation, limiting pixel availability for text-aligned sketch synthesis. We propose a potential mitigation strategy (Fig.15-right): Enhancing reference images via zooming and replication-based padding to increase pixel density. While this alleviates the issue partially, the augmentation process inevitably alters original stylistic attributes (e.g., sparsity distribution and stroke thickness). Consequently, fully resolving this limitation necessitates further investigation into non-destructive reference adaptation methods.

## I   Social Impact

As illustrated in Fig. 4 and Fig. 12, M3S enables high-quality artist-style sketch generation with an exemplar. It takes only a few tens of seconds to produce work that would take a human hours. Meanwhile, as shown in Fig. 5 and Fig. 13, our novel multi-style sketch generation technology and style preference control can provide users with more creative inspiration.

## J   Selected Style Images of Style 5

As illustrated in Fig. 16, we curated sketches spanning diverse artistic styles, including portraiture, scalar vector graphics, meticulous brushwork (gongbi), minimalist aesthetics, and detailed rendering styles, among others.

## K   Prompts for Quantitative Experiments

The textual prompts for quantitative evaluation cover a wide range of common real-life scenarios and categories, including animals, landscapes, vehicles, and daily objects etc. Specific prompts are as follows:

1. a sketch of a sailboat floating on calm water
2. a sketch of a pine tree on a small hill
3. a sketch of a cat sitting inside a teacup
4. a sketch of a hot air balloon high over mountains
5. a sketch of a dragon flying in the sky, full body
6. a sketch of a bicycle leaning against a brick wall
7. a sketch of a small house with two windows
8. a sketch of a fruit basket with fruits

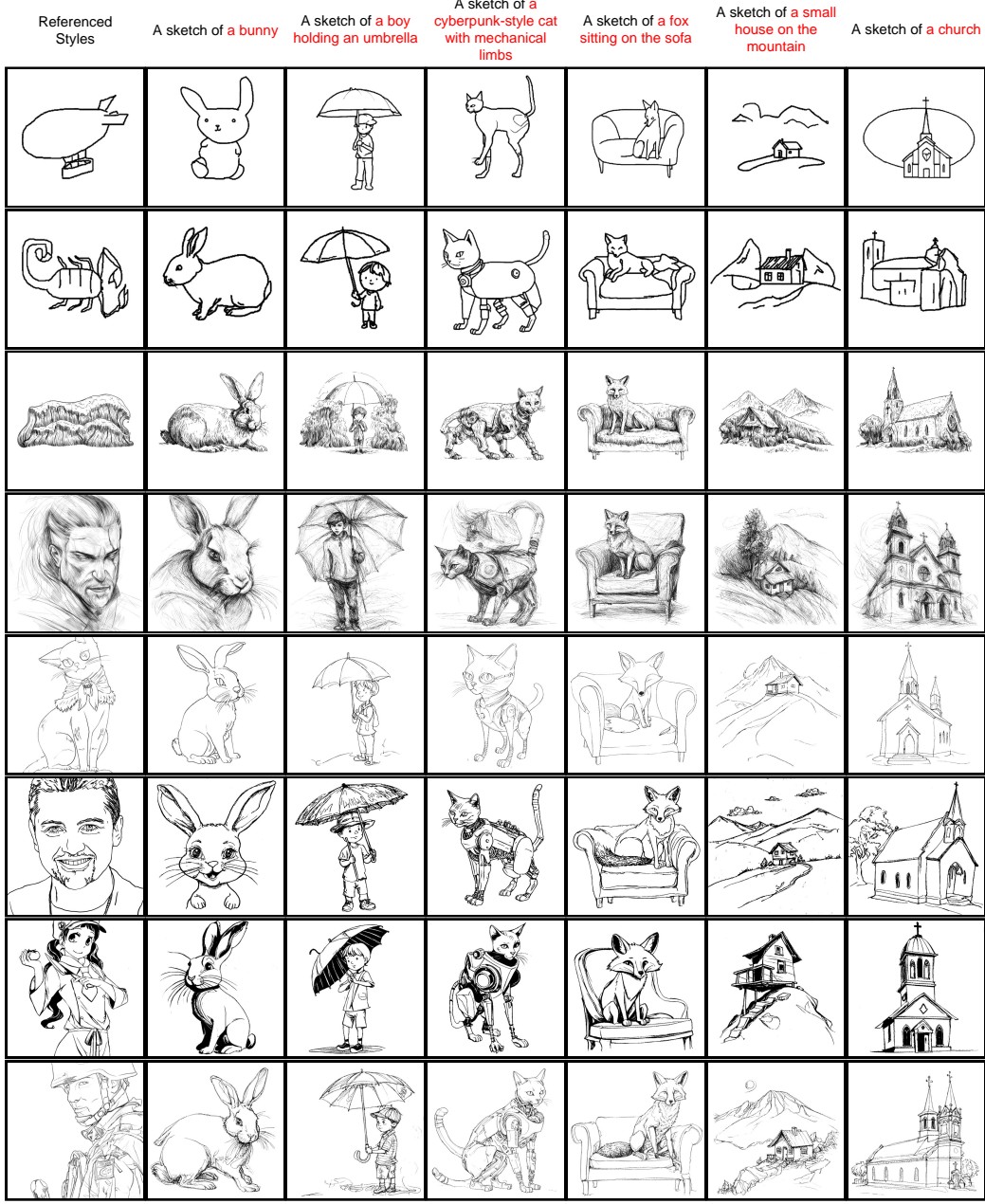

Figure 12: Sketches generated by the proposed M3S (SDXL). The results in each column are obtained using the same prompts and different referenced styles.

9. a sketch of a portrait of naruto
10. a sketch of enchanted forest with glowing mushrooms
11. a sketch of James Bond in a tuxedo and holding a gun
12. a sketch of a man performing tai chi
13. a sketch of a butterfly with plain wings
14. a sketch of a campfire under starry sky
15. a sketch of sydney opera house
16. a sketch of a bridge over a narrow river

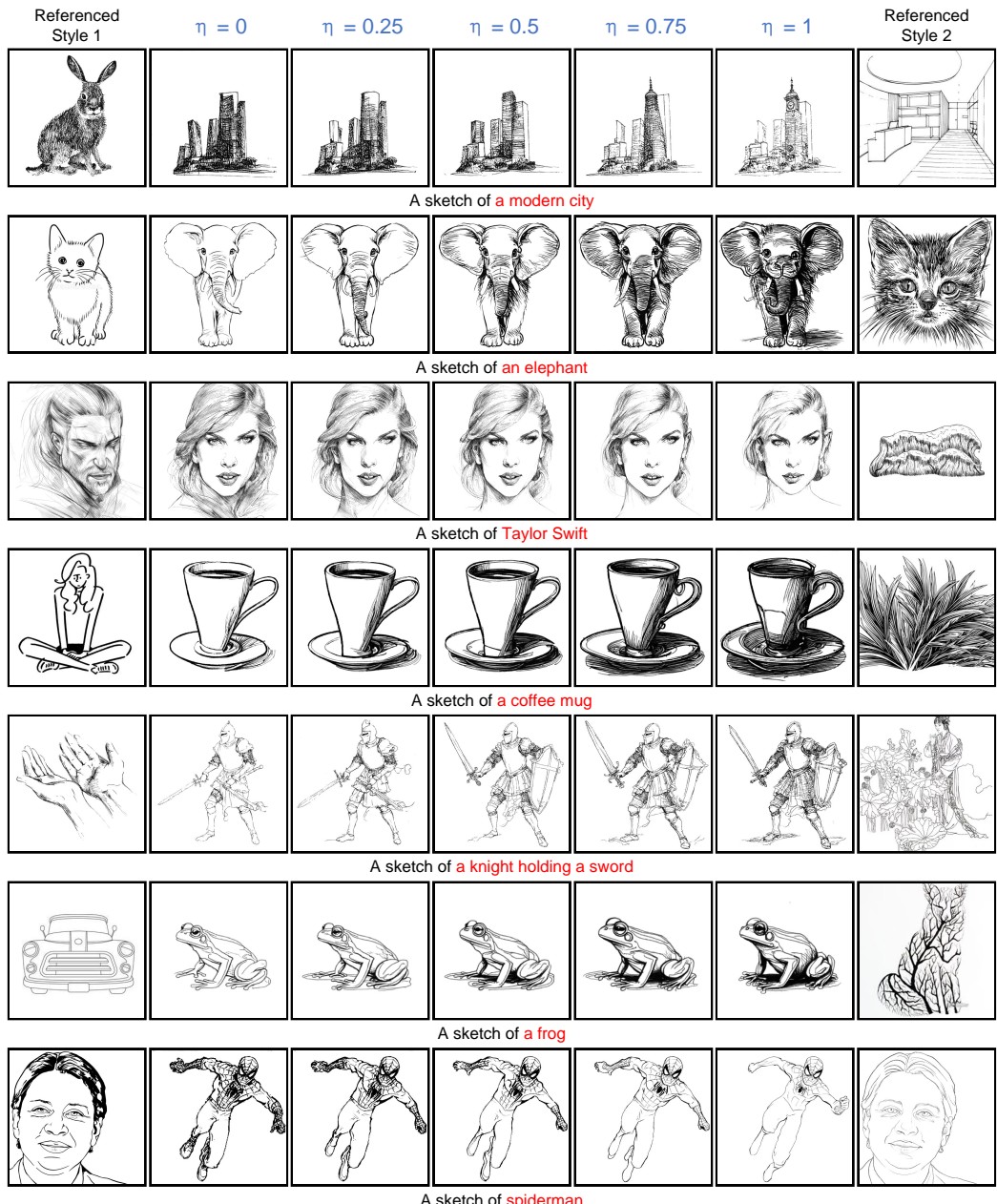

| Referenced Style 1 | η = 0 | η = 0.25 | η = 0.5 | η = 0.75 | η = 1 | Referenced Style 2 |

A sketch of a modern city

A sketch of an elephant

A sketch of Taylor Swift

A sketch of a coffee mug

A sketch of a knight holding a sword

A sketch of a frog

A sketch of spiderman

Figure 13: Generated sketches by M3S (SDXL) with multi-styles. The parameter η is used for controlling the style tendency.

17. a sketch of a robot holding a flower
18. a sketch of two white bunnies
19. a sketch of a lighthouse by the ocean
20. a sketch of an owl perched on a car
21. a sketch of a single feather falling through air
22. a sketch of a pair of glasses on an open book
23. a sketch of a steaming coffee cup on a saucer
24. a sketch of a crescent moon with one star

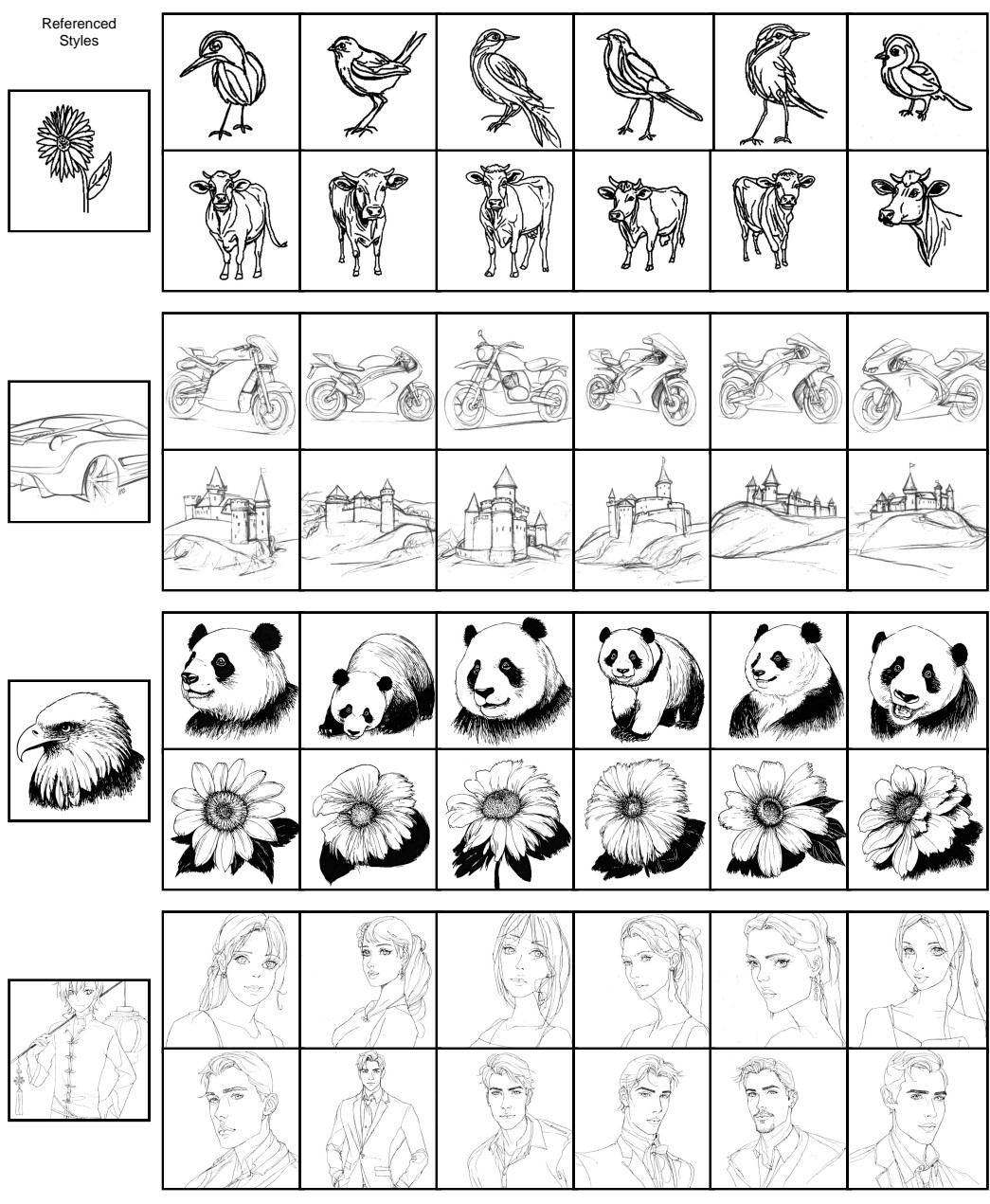

Figure 14: Generated sketches of the same styles and the same prompts in each row. We set different seeds to synthesize various sketches by M3S (SDXL).

25. a sketch of a fox standing on a log
26. a sketch of an empty swing hanging from a tree
27. a sketch of a vintage camera on a tripod
28. a sketch of a seashell on smooth sand
29. a sketch of a paper airplane mid-flight
30. a sketch of a single rose in a slim vase
31. a sketch of a mountain peak piercing clouds
32. a sketch of a deer standing in a meadow

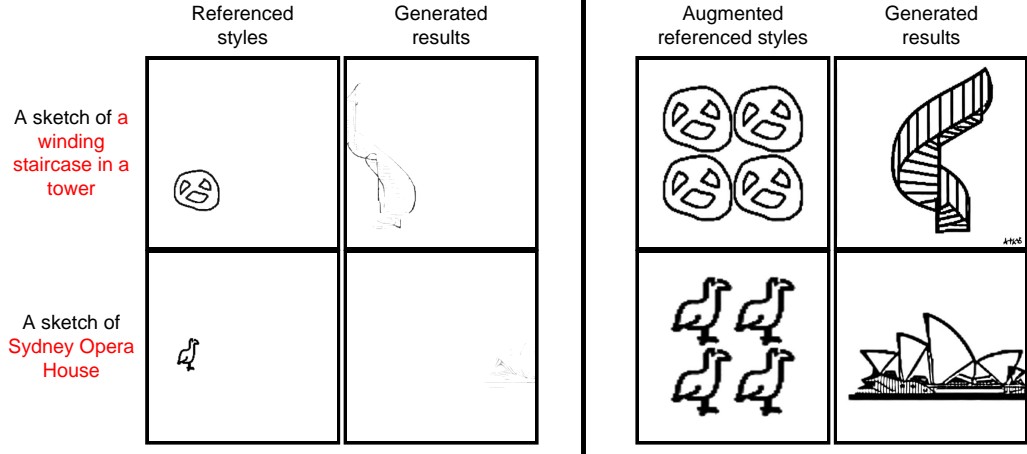

Figure 15: Left: Failure cases of M3S. When the referenced sketches are too small or sparse, M3S is difficult to produce meaningful results. Right: A potential resolution through image augmentation.

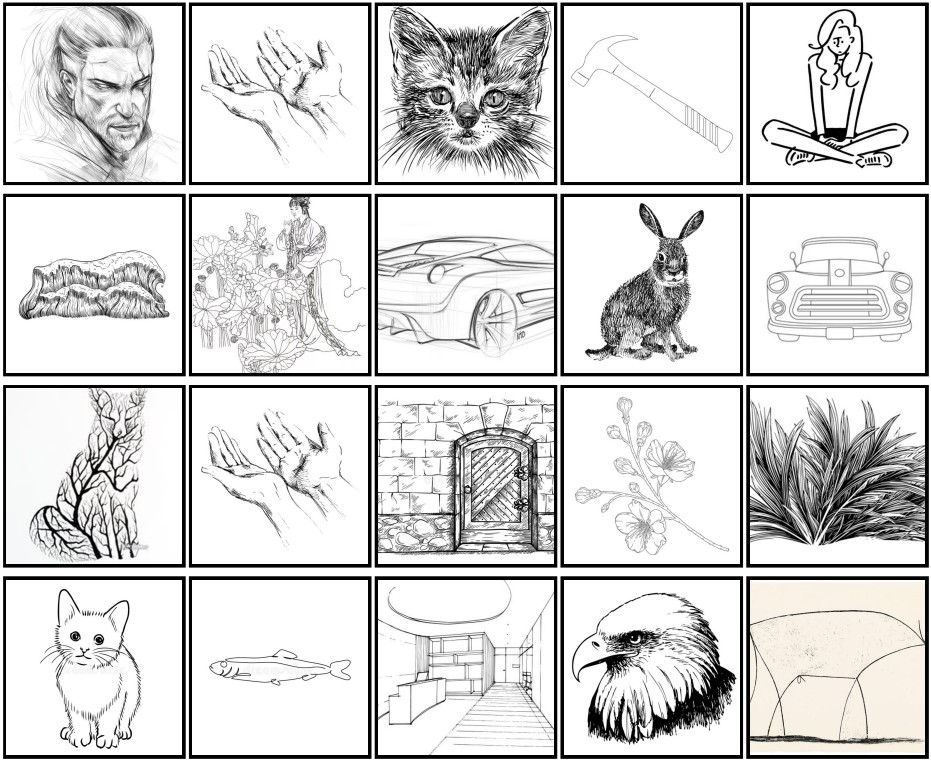

Figure 16: The selected images of style 5.

33. a sketch of a hanging lantern at night
34. a sketch of a winding road through a desert
35. a sketch of a penguin on an ice floe
36. a sketch of a key lying on a wooden table
37. a sketch of a waterfall cascading down rocks
38. a sketch of a pair of scissors cutting paper

39. a sketch of a street lamp in light fog
40. a sketch of a cactus in a clay pot
41. a sketch of a violin leaning on a chair
42. a sketch of a squirrel holding an acorn
43. a sketch of a single leaf floating on water
44. a sketch of an open umbrella against wind
45. a sketch of a stone arch bridge at dawn
46. a sketch of a chess piece on a board
47. a sketch of a crow perched on a fence
48. a sketch of a winding staircase in a tower
49. a sketch of a person doing yoga pose
50. a sketch of a winding river through hills

## L   Text Prompts for Fig. 5

The textual prompts for the top-right of Fig. 5 in each row:

1. a sketch of an apple
2. a sketch of a teddy bear
3. a sketch of iron man
4. a sketch of Albert Einstein

Top-left:

1. a sketch of Eiffel Tower
2. a sketch of a dog wearing a hat
3. a sketch of a birthday cake
4. a sketch of a coconut tree

