# OpenReview forum: "Text to Sketch Generation with Multi-Styles"
_NeurIPS.cc/2025/Conference — NeurIPS 2025 poster_

### Official Review · Reviewer_kzEp · 2025-06-29

**Clarity:** 2
**Significance:** 2
**Originality:** 2
**Rating:** 4
**Confidence:** 3

**Summary:**

The paper proposes a text-to-sketch generation method. The method use reference features as auxiliary information with linear smoothing and use a style-content guidance mechanism. The framework can be extended to handle multiple styles (interpolating between different styles). Experimental results show that the proposed method can provide better style-content disentanglement.

**Questions:**

- As mentioned in "Strengths And Weaknesses", the generation speed might be an issue. It would be good to list the break down of time usage for different modules so that the readers can better understand the bottleneck of the proposed method.

- The paper compares with many general baseline works that are designed for general styles. It would be good if the author can clearly mark which methods are designed for general styles, while which methods are designed for sketches only. This will help readers understand the scope of the baseline approaches.

- It is usually subjective to evaluate the quality of a sketch. It would be more convincing if user study results are included in the paper.

**Ethical Concerns:**

["NO or VERY MINOR ethics concerns only"]

**Final Justification:**

The rebuttal answered my questions about speed, motivation, and generalizability. Thus I am raising my rating

**Limitations:**

The authors have discussed the limitations.

**Quality:**

2

**Strengths And Weaknesses:**

Strengths:

+ The experimental results shows that the proposed text-to-sketch method outperforms existing baseline approaches, both quantitatively and qualitatively.
+ The explanation of the proposed method is clear and the paper is easy to follow.

Weaknesses:
- The main weakness is that text-to-sketch generation might be a sub-domain of text-to-image generation. It is questionable whether it is necessary to design a sophisticated method for the text-to-sketch task. The author mentioned in the paper that existing text-to-image models are not good at sketch generation, e.g., they are designed for color and natural images. However, this problem might just come from training data. If we replace training data, or finetune the existing model with additional sketch images, or train a sketch lora, this problem might have been solved. Different trigger words can be added to training captions to incorporate different sketch styles.

- The proposed method uses null-text-inversion. However, this method usually requires optimization and make the method slow. The paper mentions "Each sketch takes about 40 seconds (M3S (SD v1.5)) and 70 seconds (M3S (SDXL)) on an A100 40GB GPU", but there is not comparison with baseline approaches about generation time. The proposed method might be significantly slower than existing approaches, limiting the applications of this approach.

---

> ### Author Rebuttal · Authors · 2025-07-29
>
> # Response to Reviewer 58qS
> We thank the reviewer for the constructive comments and feedback on our paper. Regarding the concerns of the reviewer 58qS, we provide the following responses.
>
> ### **Q1. As mentioned in "Strengths And Weaknesses", the generation speed might be an issue. It would be good to list the break down of time usage for different modules so that the readers can better understand the bottleneck of the proposed method.**
>
> A1. Thanks for your valuable feedback. We employ the null-text inversion technique via **DDPM-Inversion** [a], which constructs the latent noise sequence through a forward process, computes and retains the true noise values, and directly utilizes these values during sampling to estimate the target noise—enabling perfect reconstruction. In other words, it can be considered an optimized version of DDIM-Inversion [b]. Crucially, **this is not an optimization-based approach**, thus avoiding the multi-minute optimization time required by methods like [c]. For single-style generation tasks, our method's average inference time per image is reported below:
>
>  + **M3S (SD v1.5)​: 41 seconds in total, where 6 seconds for DDPM-Inversion trajectory collection and 35 seconds for denoising synthesis.**
>
> + **M3S (SDXL)​: 63 seconds in total, where 10 seconds for DDPM-Inversion trajectory collection and 53 seconds for denoising synthesis.**
>
> The table below records the inference time per sketch generation across different models. Our M3S (SD v1.5) demonstrates slower generation than InstantStyle but comparable speed to CSGO, StyleStudio, and RB-Modulation. While M3S (SDXL) requires additional processing time, it outperforms StyleAligned and AttentionDistillation in efficiency. Notably, InstantStyle, CSGO, RB-Modulation, and StyleStudio rely on **external pre-trained models** beyond their diffusion backbones, incurring non-inference overheads (e.g., RB-Modulation's SAM [d] loading takes minutes). Although InstantStyle offers speed advantages, its dependency on IP-Adapter [e] **necessitates model-specific pre-training**. In contrast, M3S, StyleAligned, and AttentionDistillation—despite longer inference—achieve superior style consistency with **training-free, plug-and-play flexibility**.
>
> || StyleAligned | AttentionDistillation | CSGO | StyleStudio | RB-Modulation | InstantStyle | Ours (SDXL) | Ours (SD v1.5) |
> | :------- | :----------- | :------------------- | :--- | :--------- | :------------ | :----------- | :---------- | :------------- |
> | Time/Second|76| 98 | 38 |37 | 51 | 13 | 63 | 41 |
>
> ### **Q2. The paper compares with many general baseline works that are designed for general styles. It would be good if the author can clearly mark which methods are designed for general styles, while which methods are designed for sketches only. This will help readers understand the scope of the baseline approaches.**
>
> A2. Thanks for your suggestions. The baseline methods, including InstantStyle, CSGO, StyleStudio, RB-Modulation, AttentionDistillation, and StyleAligned are designed for **general scenarios**. Other methods, like SketchRNN, CLIPasso,
> and DiffSketcher are **specifically built for sketch generation**. We will clarify this categorization more explicitly in the revised version. Extensive comparisons with general-purpose methods are conducted because the scarcity of same-style sketch data, e.g., the 4skst [f] dataset contains only 25 sketches per style across 4 styles, makes it impractical to train dedicated style-specific text-to-sketch models. Addressing this data scarcity to achieve zero-shot style-specific generation is a core challenge our work tackles.
>
> ### **Q3. It is usually subjective to evaluate the quality of a sketch. It would be more convincing if user study results are included in the paper.**
>
> A3. We appreciate your constructive comments. We conducted a **​human preference assessment**​ via structured questionnaires to evaluate text-to-sketch synthesis performance. Each questionnaire contained:
> 1. Six randomly selected sets of generated sketches.
> 2. Eight anonymized outputs per set from different models under identical prompts and reference styles.
> 3. Evaluation criteria: Comprehensive assessment across three dimensions—text alignment, style consistency, and generation quality.
>
> Participants ranked results on a scale of 1-8 (8=optimal). From 58 submissions (avg. completion: 4m16s), we excluded 14 invalid responses (<60s completion/missing rankings), retaining ​44 validated questionnaires. The results are shown on the table below.
>
> Quantitative analysis of user preferences revealed:
> 1. **​M3S Dominance**:
>    + M3S (SDXL) achieved the highest average score (**6.19**) and M3S (SD v1.5) secured strong performance (**5.44**).
>    + Both variants significantly outperformed baseline methods, demonstrating excellent balance across evaluation dimensions.
>    + We evaluate the statistical significance by **rank tests**, revealing that M3S (SD v1.5) significantly outperforms all baseline methods except InstantStyle (p-value=0.26). When we align the backbone with InstantStyle (i.e., SDXL), our M3S (SDXL) demonstrates superiority over it **(p-value=1.06e-5)**.
> 2. **Baseline Comparisons**:
>    + ​InstantStyle​ attained a competitive score (5.08) due to high aesthetic quality despite style consistency limitations.
>    + ​StyleAligned​'s style advantages were compromised by content leakage issues.
>    + AttentionDistillation​ achieved marginal improvements over StyleStudio/RB-Modulation through style-focused optimization, notwithstanding quality deficiencies.
>
> In conclusion, the study affirms M3S's superior alignment with human evaluator preferences. We will include the results and analysis in the revision version.
>
>
> ||StyleAligned|AttentionDistillation|CSGO|StyleStudio| RB-Modulation|InstantStyle|Ours (SDXL)|Ours (SD v1.5)|
> | :------- | :----------- | :------------------- | :--- | :--------- | :------------ | :----------- | :---------- | :------------- |
> |Rating| 2.77|4.28|3.83|4.22|4.20|5.08|**6.19**|5.44|
>
> ### **Q4. The main weakness is that text-to-sketch generation might be a sub-domain of text-to-image generation. It is questionable whether it is necessary to design a sophisticated method for the text-to-sketch task. The author mentioned in the paper that existing text-to-image models are not good at sketch generation, e.g., they are designed for color and natural images. However, this problem might just come from training data. If we replace training data, or finetune the existing model with additional sketch images, or train a sketch lora, this problem might have been solved. Different trigger words can be added to training captions to incorporate different sketch styles.**
>
> A4. Thank you for your suggestion for style-specific sketch generation solutions. For specific-style sketch generation, training-based methods face three major challenges in practical applications:
>
> 1. **Data Scarcity and Annotation Difficulties**
>    Professionally styled sketch datasets are **scarce and typically small-scale** (e.g., 25 sketches per style in 4skst [f], 140 in APDrawing [g]). Effective fine-tuning requires dozens of samples per style, necessitating thousands of sketches for diverse styles like our Style 5 dataset's 20 distinct categories. Although web-sourced sketches are plentiful, style annotation poses significant hurdles: multimodal models (e.g., BLIP-2 [h]) trained on real image-language pairs will struggle with sketch style categorization. **Manual annotation is costly and inconsistent** due to sketches' abstract nature—style descriptors vary across annotators' age, cultural background, and regional context. Such inaccuracies impede models from learning precise token representations for style triggering.
>
> 2. **​Limited Flexibility in Trained Approaches**
>    Training-based methods require model-specific adaptations, whereas our training-free technique transfers seamlessly across architectures. Furthermore, generating novel styles absent from training data necessitates re-training, while M3S operates zero-shot without this constraint. Critically, our framework inherently supports multi-style fusion, whereas trained methods lack this capability without extensive additional research.
>
> 3. **​The "An Image is Worth One Word" Principle [i]**
>    Given the linguistic ambiguity of sketch styles, users must memorize dozens to hundreds of trigger tokens to match target styles accurately. M3S eliminates this burden by using a single reference image for style guidance, significantly enhancing usability.
>
> Overall, our approach circumvents the data acquisition constraints and flexibility limitations inherent in training-based methods while enabling novel multi-style fused sketch generation. We appreciate your valuable suggestion regarding training-based alternatives and will investigate these directions in future work.
>
> [a] Huberman-Spiegelglas, et al. "An edit friendly ddpm noise space: Inversion and manipulations." CVPR 2024.
>
> [b] Song, Jiaming, et al. "Denoising diffusion implicit models." arXiv preprint arXiv 2020.
>
> [c] Mokady, Ron, et al. "Null-text inversion for editing real images using guided diffusion models." CVPR 2023.
>
> [d] Kirillov, Alexander, et al. "Segment anything." ICCV 2023.
>
> [e] Ye, Hu, et al. "Ip-adapter: Text compatible image prompt adapter for text-to-image diffusion models." arXiv 2023.
>
> [f] Seo, Chang Wook, Amirsaman Ashtari, and Junyong Noh. "Semi-supervised reference-based sketch extraction using a contrastive learning framework." TOG 2023: 1-12.
>
> [g] Yi, Ran, et al. "Apdrawinggan: Generating artistic portrait drawings from face photos with hierarchical gans." CVPR 2019.
>
> [h] Li, Junnan, et al. "Blip-2: Bootstrapping  language-image pre-training with frozen image encoders and large language models." ICML 2023.
>
> [i] Gal, Rinon, et al. "An image is worth one word: Personalizing text-to-image generation using textual inversion." arxiv 2022.

---

> > ### Comment · Reviewer_kzEp · 2025-08-05
> >
> > Thanks for the detailed response. The rebuttal (especially the user study) has resolved most of my concerns. Thus I am raising my rating.

---

### Official Review · Reviewer_W249 · 2025-07-02

**Clarity:** 3
**Significance:** 2
**Originality:** 2
**Rating:** 5
**Confidence:** 4

**Summary:**

This paper proposes a method called M3S for generating stylized sketches from a text prompt, guided by one or two reference sketches. Instead of training a new model, the authors use a pre-trained Stable Diffusion model and inject style features by modifying the key (K) and value (V) components in the attention layers of the pre-trained model. They also leverage an AdaIN-based control mechanism that enables users to interpolate between different sketch styles. They propose a dual guidance formulation that separates text (content) and style impact, allowing for more fine-grained control over the generated sketches. Quantitative results suggest partial improvement over existing methods. This model also supports multi-style blending, which can be the unique feature of this model.

**Questions:**

To strengthen the paper, I recommend the following improvements:

**1. [Conducting a user study]:** Due to the subjectivity of sketch aesthetics, conducting a user study could help to verify this method superiority to existing baselines.

**2. [Handling diverse reference styles]:** I would like to see how this model handles interpolating between styles, especially when two styles are really different from each other.

**3. [Eq. 5 might be better represented]:** I think authors might improve the representation of Eq. 5. After reading the text, I could figure it out but it should somehow be more clear in the equation (somehow it should show where blended K/V features are used).

**4. [Failure cases]:** It will be great if authors mention which kind of styles result in failure of this model. For example, can this model imitate the style of pointillism sketches (sketches drawn by many points)?

**5. [Few missing references]:** Some reference-based sketch extraction studies from the computer graphic domain might be missing (see below). This is not a main concern; and I trust the authors to add if they found new ones; if not, it was a very minor comment that they can skip it. Here I added examples of missing studies that introduced style transfer concept into the sketch domain: [1] SketchPatch: Sketch stylization via seamless patch-level synthesis. TOG 2020, [2] Reference Based Sketch Extraction via Attention Mechanism, TOG 2022.

I am on the positive side for this paper. If authors clearly demonstrate that their method is outperforming existing baselines or addressing some of the main concerns (like human evaluation), I am willing to increase the score after reading the author comments.

**Ethical Concerns:**

["NO or VERY MINOR ethics concerns only"]

**Final Justification:**

Thanks for authors for addressing my concerns. I raised my initial positive score.

**Limitations:**

It can be improved by showing failure cases, as it was explained in the review.

**Paper Formatting Concerns:**

I did not see any major concerns.

**Quality:**

3

**Strengths And Weaknesses:**

Thanks for the authors effort to prepare the manuscript to advance the reference-based sketch extraction domain. Here, I listed the strengths and room for improvement:

**- Strengths:**

**1. [Simple yet effective model]:** I like the simplicity of the proposed method. It does not require re-training and it is built upon the existing knowledge in feature fusion and blending. It uses existing methods (e.g., K/V blending, AdaIn, and dual guidance) to merge style and content features in a smart way.

**2. [Simple control over generated sketch for a human user]:** From a user point of view, the method seems to give lots of freedom in the generated sketch. Users can easily define two reference styles and control the generated sketch to be similar to one of those styles or to even have in between styles.

---------------

**- Room for improvement:**

**1. [Lack of a user study]:** Sketches are sparse and sometimes quantitative metrics might not be the best representative of human opinion and evaluation. I believe a user study might be needed to show how much this method outperforms existing baselines (especially considering that in some quantitative metrics, this method seems to not be the best performing one).

**2. [Limited technical novelty]:** The proposed method builds on many existing ideas in the literature, which could be seen as a limited technical novelty. However, if the model clearly outperforms previous baselines, I would consider this a minor issue. For example, AdaIN blending has long been used for arbitrary style transfer. K/V injection (substitution) in diffusion attention layers has been done before and here the feature fusion is improved to avoid artifacts.

**3. [Quantitative evaluation of multi-style interpolation]:** While style blending is demonstrated visually, there are no metrics provided to quantify the quality of the style interpolations; such as the smoothness of transitions or the similarity of interpolated sketches to either reference (ref1 or ref2). It is also important to evaluate the consistency of interpolation quality across different pairs of reference styles, especially when the styles differ significantly from each other.

---

> ### Author Rebuttal · Authors · 2025-07-29
>
> # Response to Reviewer W249
>
> We thank the reviewer for the constructive comments and positive feedback on our paper. Regarding the concerns of the reviewer W249, we provide the following responses.
>
> ### **Q1. Conducting a user study.**
>
> A1. We appreciate the reviewer's constructive comments. We have conducted a **user study** employing a questionnaire-based methodology to assess human preference. Each questionnaire included six randomly selected sets of generated results. Every set contained eight outputs produced by different models using identical prompts and reference styles. Participants were instructed to rank all eight anonymized results per set based on **text alignment, style consistency, and generation quality**. The user rankings for each of the eight models are converted to scores from 1 to 8 (8=best) correspondingly. We collected 58 questionnaires with an average completion time of 4 minutes and 16 seconds. After excluding 14 invalid submissions with completion time under 60 seconds and missing rankings, 44 valid responses remained. Average scores per question for each method are tabulated below.
>
> In the user study, M3S (SDXL) achieved an average score of **6.19** while M3S (SD v1.5) scored **5.44**. We evaluate the statistical significance by **rank tests**, revealing that M3S (SD v1.5) significantly outperforms all baseline methods except InstantStyle (p-value=0.26). When we align the backbone with InstantStyle (i.e., SDXL), our M3S (SDXL) demonstrates superiority over it **(p-value=1.06e-5)**. This demonstrates M3S's superior balance across style alignment, text fidelity, and generation quality. Despite InstantStyle's relatively weaker style consistency, its high aesthetic quality yielded a competitive score of 5.08. Conversely, StyleAligned's advantages in style consistency were undermined by content leakage issues. AttentionDistillation achieved marginal score improvements despite inferior generation quality compared to StyleStudio and RB-Modulation, primarily due to its style consistency. The results affirm M3S's alignment with human preferences.
>
> || StyleAligned | AttentionDistillation | CSGO | StyleStudio | RB-Modulation | InstantStyle | Ours (SDXL) | Ours (SD v1.5) |
> | :------- | :----------- | :------------------- | :--- | :--------- | :------------ | :----------- | :---------- | :------------- |
> | Rating| 2.77| 4.28 | 3.83 | 4.22 | 4.20 | 5.08 | **6.19**| 5.44 |
>
> ### **Q2. Handling diverse reference styles and quantitative evaluation of multi-style interpolation.**
>
> A2. Thanks for your valuable feedback. Following your suggestion, we have added supplementary experiments for multi-style generation. Using the same 50 prompts as the manuscript, we generated outputs for each prompt by randomly selecting two reference sketches from the Style 5 (S5) dataset. To specifically validate generation performance with significantly distinct reference styles, we conducted an additional experiment set pairing one randomly selected S5 image with one randomly chosen image from the QuickDraw (QD) dataset per prompt. **It is noted that the styles of the QD sketches are very different from those in S5.** The quantitative results are provided in the table below.
>
> | M3S implementations|Reference style| |$\eta=0$ |    | |$\eta=0.25$  | | |$\eta=0.5$ ||
> | :-------------- | :---- | :------------ | :-------- | :-------- | :------------ | :-------- | :-------- | :------------ | :-------- | :-------- |
> | | | CLIP-T(↑)     | DINO-ref1   | DINO-ref2 | CLIP-T | DINO-ref1| DINO-ref2 | CLIP-T | DINO-ref1|DINO-ref2|
> |  SDXL | S5-S5 |0.3442| 0.3936 | 0.4944| 0.3514|0.4180 | 0.4821 |0.3495 |0.4408|0.4556  |
> |SD v1.5 |S5-S5| 0.3465| 0.3850 | 0.4776 | 0.3453|0.4215| 0.4597|0.3499|0.4469 |0.4509|
> | SDXL | QD-S5 |0.3426|0.3051|0.4724|  0.3455 | 0.3266|0.4622|0.3457 |0.3330|0.4397|
> |SD v1.5 |QD-S5|0.3434| 0.3630|0.4339| 0.3417|0.3948 |0.4236|0.3452 |0.4102|0.4057|
>
> |M3S implementations|Reference style||$\eta=0.75$|||$\eta=1$||
> |:-------------- |:---- | :------------ | :-------- | :-------- | :------------ | :-------- | :-------- |
> |||CLIP-T(↑)|DINO-ref1 |DINO-ref2|CLIP-T(↑)|DINO-ref1|DINO-ref2|
> |SDXL|S5-S5|0.3499|0.4578|0.4221|0.3470|0.4693|0.3975|
> |SD v1.5|S5-S5|0.3478| 0.4528|0.4257|0.3528|0.4626|0.3825|
> |SDXL|QD-S5|0.3447|0.3409|0.4209|0.3396|0.3617|0.3916|
> |SD v1.5|QD-S5|0.3440|0.4250|0.3938|0.3468|0.4381|0.3766|
>
> When the references exclusively originate from S5, M3S maintains text alignment comparable to single-style generation (CLIP-T: M3S (SDXL) 0.3467, M3S (SD v1.5) 0.3494). For style consistency, DINO-ref1 exhibits a positive correlation with $\eta$ (i.e., increasing $\eta$ enhances alignment with reference 1), while DINO-ref2 shows a negative correlation. This monotonic relationship aligns with qualitative results in Figs. 1 and 5. At boundary conditions ($\eta$=0 or 1) of multi-style sketch generation, only one style participates in AdaIN modulation for image generation. Compared to single-style synthesis using that style alone, the DINO metric remains relatively ​lower​ due to persistent multi-style feature injection (**see Response to Reviewer cQmr Q1** for more analysis).
>
> For the two reference styles, replacing the pair of S5-S5 with QD-S5 reduces style consistency for both implementations (i.e., SD v1.5 and SDXL), though M3S (SD v1.5) demonstrates superior robustness. Crucially, M3S (SDXL) struggles to effectively utilize QD's abstract features, evidenced by significantly lower scores of DINO-ref1 than DINO-ref2 in QD-S5 pairs. This limitation stems from SDXL's high-fidelity optimization [a] - its user-tested superiority over SD v1.5 creates inherent incompatibility with low-quality, abstract datasets like QD. Due to OpenReview's restrictions on image uploads during the discussion phase, we cannot directly provide corresponding visual results. However, the overall generation quality is visually analogous to the butterfly reference outputs in Fig. 5 of the main paper.
>
> ### **Q3. Eq. 5 might be better represented.**
>
> A3. We appreciate your suggestion. We will revise Eq. (5) to explicitly represent the $K/V$ injection mechanism as follows. Our formulation follows the notation established in [b]. We will also add explanatory text regarding the equation and cite [b] to enhance readability.
>
> $\tilde{\epsilon}\_t=\epsilon\_{\theta}(\textbf{z}\_t^{tar},t,\emptyset)+\omega\_1\underbrace{(\epsilon\_\theta^\times(\textbf{z}\_t^{tar},t,text,K^{ref},V^{ref}) -\epsilon\_\theta(\textbf{z}\_t^{tar},t,\emptyset ))}\_{content\ guidance\ direction} + \omega\_2 \underbrace{(\epsilon\_\theta^\times (\textbf{z}\_t^{tar},t,\emptyset,K^{ref},V^{ref}) -\epsilon\_\theta(\textbf{z}\_t^{tar},t,\emptyset ))}\_{style\ guidance \ direction}$
>
>
> ### **Q4. Failure cases.**
>
> A4. We apologize for placing the failure cases in an insufficiently prominent location (specifically, **Appendix H and Fig. 15** of manuscript). In the upcoming revision, we will incorporate these directly into the main body. Our method's primary failure mode occurs when reference images are excessively sparse with content concentrated in small regions, potentially yielding near-blank outputs. To address this, we introduce a solution in Appendix H as follows, i.e., enhancing reference images via zooming and replication-based padding to increase pixel density.
>
> Following your suggestion, we tested our method using Google-collected pointillism sketches (depicting flowers, balls, tigers, etc.). Our method succeeded in most examples. However, we observed **one failure case** where **SD's VAE failed to accurately reconstruct the reference image**, indicating inaccurate latent space representations. This example will be incorporated into the failure cases section in the revised manuscript.
>
> ### **Q5. Few missing references.**
>
> A5. We appreciate your feedback on the missing literature. In the revised manuscript, we will incorporate the referenced computer graphics papers relevant to our work and conduct an additional literature review to provide readers with a more comprehensive domain context.
>
> ### **Q6. Limited technical novelty.**
>
> A6. Thanks for your constructive feedback. While building upon prior concepts, M3S extends existing techniques in novel directions to address unique challenges in **​multi-style sketch synthesis**:
>
> 1. **Multi-reference feature fusion**
>    Prior K/V injection methods (e.g., [b,c,d]) focused on single-style transfer. M3S introduces: **Concurrent injection​ of features** from multiple references (Section 3.1) and blending them through linear smoothing.
>
> 2. **Adaptive noise modulation**
>    Traditional AdaIN [e] achieves style transfer by modulating image features, but its outputs remain constrained by the content image. Recent diffusion-based methods [b] primarily focus on artifact removal without exploring AdaIN's style modulation capabilities. In contrast, **M3S demonstrates that multi-style fusion can be achieved by applying AdaIN to denoising trajectories in noise space, free from encoded-space limitations**. Furthermore, M3S leverages diffusion models' generative capacity to automatically **produce creative content** under varying $\eta$ conditions, such as the evolving beard details in Messi's portrait (Fig. 5).
> 3. **Performance Validation**
>    M3S demonstrates consistent advantages in human evaluations, which achieves highest user preference score (6.19​ for SDXL).
>
> In summary, our core innovation centers on both the novel task of multi-style sketch synthesis and its corresponding framework design. Building on this framework, we will further explore the feasibility of multi-style fusion generation in RGB space.
>
>
> [a] Podell, Dustin, et al. "SDXL ...." arXiv 2023.
>
> [b] Alaluf, Yuval, et al. "Cross-image attention ..." SIGGRAPH 2024.
>
> [c] Hertz, Amir, et al. "Style aligned ..." CVPR 2024.
>
> [d] Cao, Mingdeng, et al. "Masactrl ..." ICCV 2023.
>
> [e] Huang, Xun, et al. "Arbitrary style transfer ..." ICCV 2017.

---

> > ### Comment · Reviewer_W249 · 2025-08-06
> >
> > The rebuttal addresses my main concerns by adding a user study with statistical results, providing quantitative evaluation for multi-style interpolation with diverse and distinct reference styles, and clarifying Eq. 5. With these clarifications, I remain positive toward acceptance and increase my initial positive score.

---

### Official Review · Reviewer_cQmr · 2025-07-02

**Clarity:** 3
**Significance:** 3
**Originality:** 3
**Rating:** 4
**Confidence:** 5

**Summary:**

This paper presents a novel training - free diffusion model framework for multiple - style references. By integrating reference features as auxiliary information through linear smoothing, it effectively suppresses reference - sketch content leakage. Meanwhile, via a joint AdaIN module for coordinating multi - style generation, it achieves flexible style control without large - scale training, enhancing synthesis quality and expanding diversity.

**Questions:**

This method has certain bias when processing certain objects. As shown in the clown example in Figure 1, even with $\eta$ set to 0, the generated clown image doesn't have a simple drawing style, though clowns are suitable for simple drawing. This echoes the fact that the method doesn't achieve optimal results in all cases in Table 2. What's the deep reason for this?

The paper only shows results of referring to multiple styles, but the used reference sketches are all of high - quality. So when encountering cases with big differences in style and drawing quality, the method's performance is unclear. For example, what's the effect when using amateur sketches (such as those in QuickDraw) and professional sketches as style references?

**Ethical Concerns:**

["NO or VERY MINOR ethics concerns only"]

**Limitations:**

Referring to the suggestions put forward in the Questions

**Quality:**

3

**Strengths And Weaknesses:**

This paper presents a novel training - free diffusion - model - based framework. It can combine textual prompts and reference - style sketches to achieve accurate style guidance. Experiments show its significant advantages in style - alignment accuracy and control flexibility. The paper is well - structured, with clear method descriptions and detailed arguments. However, the need to adjust many hyperparameters restricts its practical - application convenience.

---

> ### Author Rebuttal · Authors · 2025-07-29
>
> # Response to Reviewer cQmr
>
> We thank the reviewer for the constructive comments and positive feedback on our paper. Regarding the concerns of the reviewer cQmr, we provide the following responses.
>
> ### **Q1. This method has certain bias when processing certain objects. As shown in the clown example in Figure 1, even with $\eta$ set to 0, the generated clown image doesn't have a simple drawing style, though clowns are suitable for simple drawing. This echoes the fact that the method doesn't achieve optimal results in all cases in Table 2. What's the deep reason for this?**
>
> A1. We thank the reviewer for the valuable feedback. When $η=0$, M3S synthesizes sketches that **​blend both reference styles​ rather than adopting a single style** in this case, while strategically biasing the output toward the aesthetic attributes of Reference Style 1.  We explain this phenomenon through M3S' three core components:
>
> + **K/V Injection**: The clown example involves multi-style generation where K/V features from both reference images are injected. The diffusion model automatically assigns contour and texture to different objects. At $\eta=0$, the unicycle exhibits pure freehand sketch style, and the clown maintains freehand-style contours but develops body textures. This occurs because Stable Diffusion's natural image training data contains more detailed clown depictions (colorful costumes, facial makeup) than unicycles. Consequently, the model prioritizes texture-rich styles for clown details.
>
> + **Joint AdaIN Module**: AdaIN modulates pixel/feature distributions statistically rather than enforcing explicit structural constraints. Thus, at $\eta=0$, the result shows global distribution aligns with freehand sketch style and local discrepancies persist in structural details.
>
> + **Style-Content Guidance**: The phenomenon arises from the interplay between reference style characteristics and guidance scaling. When using sparse (cup) and dense (wave) references, the wave’s rich textures enhance content generation due to their structural relevance. For the clown example, however, the default setting of $\omega_1 = 15, \omega_2 = 15$ amplifies texture density from the non-freehand reference style. This occurs because higher $\omega_2$ prioritizes style alignment, favoring denser patterns. When we reduce $\omega_2$ to 7.5, the textures for the clown are more line-dominant (closer to freehand). Since OpenReview does not allow to upload images during the discussion period, we recommend that the reviewer verify the results via the code and dataset in the Supplementary materials.
>
> In essence, this phenomenon reflects the model's automatic selection of multi-style features for category-specific synthesis. Since we inject features from multiple references, the generated image inherently blends diverse styles even at $\eta=0$. **To achieve pure freehand sketch style, we recommend using sparse references (e.g., the cup) as the sole style source.**
>
> ### **Q2. The paper only shows results of referring to multiple styles, but the used reference sketches are all of high - quality. So when encountering cases with big differences in style and drawing quality, the method's performance is unclear. For example, what's the effect when using amateur sketches (such as those in QuickDraw) and professional sketches as style references?**
>
> A2. We appreciate the reviewer's constructive comments. To verify generation quality when using **two visually distinct reference styles**, we experimented with the following design. We first constructed a style dataset by randomly selecting 50 images from the **low quality dataset QuickDraw (QD)**, then generated 50 multi-style images using prompts identical to our main quantitative experiments. Specifically, we select one random reference from QD and one from **high quality dataset Style 5 (S5)** per generation. To contrast results against cases where both references exhibit high quality, we generated additional outputs using exclusively S5 references. Quantitative results are provided in the table below.
>
> The results indicate that QD-S5 and S5-S5 achieve comparable CLIP-T scores with only marginal degradation, demonstrating our method's robust text alignment even with significantly divergent reference styles. For SD v1.5, DINO metrics show only minor deterioration when using QD references. However, QD-S5 SDXL exhibits substantially reduced DINO-ref1, indicating the model's stronger preference for high-quality S5-aligned outputs over QD. This occurs because SDXL is explicitly optimized for high-fidelity generation and significantly outperforms SD v1.5 in user evaluations [a], making it more susceptible to quality discrepancies in reference images. Consequently, M3S (SD v1.5) is preferable for multi-style generation involving references with major quality differences.
>
> | M3S implementations|Reference style |         |$\eta=0$  |           |         |$\eta=0.25$        |           |         |$\eta=0.5$         |           |
> | :-------------- | :---- | :------------ | :-------- | :-------- | :------------ | :-------- | :-------- | :------------ | :-------- | :-------- |
> |                 |       | CLIP-T(↑)     | DINO-ref1   | DINO-ref2 | CLIP-T | DINO-ref1| DINO-ref2 | CLIP-T | DINO-ref1|DINO-ref2|
> | SDXL | S5-S5 |0.3442   | 0.3936    | 0.4944   | 0.3514    |0.4180   | 0.4821   |0.3495   |0.4408    |0.4556  |
> |SDXL| QD-S5|  0.3426   | 0.3051 | 0.4724   |  0.3455 | 0.3266    |0.4622     |0.3457       |0.3330   |0.4397    |
> |SD v1.5|  S5-S5 | 0.3465    | 0.3850    | 0.4776 | 0.3453    | 0.4215    | 0.4597   |0.3499  |0.4469    |0.4509|
> |SD v1.5|QD-S5| 0.3434        | 0.3630   |0.4339   | 0.3417       |0.3948    |0.4236    |0.3452        |0.4102   |0.4057    |
>
> | M3S implementations|Reference style| |$\eta=0.75$          |           |         |$\eta=1$          |           |
> | :-------------- | :---- | :------------ | :-------- | :-------- | :------------ | :-------- | :-------- |
> |                 |       | CLIP-T(↑)     | DINO-ref1 | DINO-ref2| CLIP-T(↑)  | DINO-ref1 |DINO-ref2|
> | SDXL | S5-S5 |0.3499    | 0.4578|0.4221   | 0.3470   | 0.4693    | 0.3975  |
> | SDXL| QD-S5|    0.3447       | 0.3409    |0.4209 |0.3396       | 0.3617    |0.3916 |
> | SD v1.5|  S5-S5 | 0.3478    | 0.4528|0.4257 | 0.3528 | 0.4626    |0.3825  |
> |SD v1.5|QD-S5| 0.3440 | 0.4250    |0.3938   | 0.3468   |0.4381 |0.3766 |
>
>
> ### **Q3. However, the need to adjust many hyperparameters restricts its practical application convenience.**
>
> A3. Thanks for your thoughtful comments. Since human evaluation of text-conditioned sketch generation typically requires multi-factor considerations, style-specific generation frameworks commonly offer multiple parameter values for user customization [b,c]. Similar to the literature, our method also has tunable hyperparameters. We provide default parameter values for both professional and abstract sketches to satisfy user requirements, as evidenced by the user study (**See Response to W249, Q1**).
>
> [a] Podell, Dustin, et al. "Sdxl: Improving latent diffusion models for high-resolution image synthesis." arXiv 2023.
>
> [b] Lei, Mingkun, et al. "StyleStudio: Text-Driven Style Transfer with Selective Control of Style Elements." CVPR 2025.
>
> [c] Zhou, Yang, et al. "Attention distillation: A unified approach to visual characteristics transfer." CVPR 2025.

---

### Official Review · Reviewer_58qS · 2025-07-04

**Clarity:** 4
**Significance:** 2
**Originality:** 3
**Rating:** 5
**Confidence:** 2

**Summary:**

The paper presents a training-free framework for style transfer for sketch generation (image of a sketch, rather than a vector), including the ability to interpolate or use multiple reference styles. The framework is applicable for pretrained diffusion models, and the results are shown on Stable Diffusion / SDXL.

The core of the proposed technique is to, instead of using the K and V from the style source in the attention mechanism during the generation of the target image, to concatenate the style K and V matrices to the target t ones (potentially after linearly blending them with the target ones. For blending multiple styles, the linear combination of the latent images is used during the generation process.

Authors perform evaluation based on the style source images in Sketchy and 4skst. The textual prompts are generated using DeepSeek. CLIP-T score is used to measure alignment between prompt and generated images, and the distance in the space of DINO/VGG features is used to measure the style alignment with the reference style image.

**Questions:**

While the general motivation for introducing the content and style guidance losses (eq 5 is clear), the motivation to linearly increase the weight of the style guidance through the generation process is not clear. Why should this be done and how sensitive is the method to the schedule?

For the SDXL Implementation, the authors mention that style injection is important for the last layers, but injection into all layers compromises text alignment. Is there a principled way for deciding which layers to use, or some motivating ablations around it?

**Ethical Concerns:**

["NO or VERY MINOR ethics concerns only"]

**Final Justification:**

The author rebuttal addressed the style guidance losses and style injection layers. I maintain my positive rating (accept).

**Limitations:**

Yes - the paper addresses many potential issues, and acknowledges a limitation when dealing with extremely sparse reference sketches where content is concentrated in a small area.

**Paper Formatting Concerns:**

--

**Quality:**

3

**Strengths And Weaknesses:**

Quality & Clarity: Authors clearly motivate their approach and discuss techniques that didn't work (ex. the use of AdaIN from Style aligned paper in lines 139-140). They perform the ablation study on the proposed modifications and do extensive comparison with state-of-the-art, outperforming it in the majority of cases. The writing is clear and there are plenty of examples in the paper. Supplementary material contains the code necessary for reproducibility for both SDXL and Stable Diffusion 1.5, as well as the source style images.

Significance & Originality: The proposed technique is novel and well-motivated for the application domain (which is somewhat narrow, as some of the assumptions made by the authors rely, for example, on images being grayscale).

---

> ### Author Rebuttal · Authors · 2025-07-29
>
> # Response to Reviewer 58qS
>
> We sincerely thank the Reviewer for your thoughtful feedback and positive assessment of our work. We answer specific questions below.
>
> ### **Q1. While the general motivation for introducing the content and style guidance losses (eq 5 is clear), the motivation to linearly increase the weight of the style guidance through the generation process is not clear. Why should this be done and how sensitive is the method to the schedule?**
>
> A1. We thank the reviewer for the valuable feedback. We linearly increase the style guidance weight during denoising to account for diffusion models' temporal dynamics in generation [a]. Early denoising primarily establishes image structure, while later stages focus on texture refinement. Applying strong style guidance $\omega_2$ prematurely could disrupt structural integrity due to the absence of textual alignment, i.e., $(\epsilon\_\theta^\times ({z}\_t^{tar},t,\emptyset ) -\epsilon\_\theta({z}\_t^{tar},t,\emptyset ))$. For example, when style guidance is not properly scheduled, the prompt "a sketch of a monkey eating a peach" might activate the generation of connecting the arm directly to the peach. Quantitative results for Style 1-5 generations using M3S (SDXL) are provided in the Table below. Without linear scheduling, CLIP-T and aesthetic scores decrease while the DINO score increases. This occurs because without linear scheduling, the generation process over-prioritizes style consistency. However, higher style consistency doesn't guarantee better visual quality and may cause severe unnatural artifacts in local content. We therefore employ linear scheduling to **balance generation quality, text alignment, and style consistency**. This technique also implemented in [b].
>
> | Method          |       |         |Style1           |           |         |Style2           |           |         |Style3           |           |
> | :-------------- | :---- | :------------ | :-------- | :-------- | :------------ | :-------- | :-------- | :------------ | :-------- | :-------- |
> |                 |       | CLIP-T(↑)     | DINO(↑)   | asesthetic    | CLIP-T(↑)     | DINO(↑)   | asesthetic    | CLIP-T(↑)     | DINO(↑)   | asesthetic    |
> | With Linear     |       | 0.3607    | 0.6545    | 4.9802 | 0.3556    | 0.6531    | 5.0581   |0.3422 | 0.6041    |5.1605|
> | Without Linear  |       | 0.3571        | 0.6813    |4.8622   | 03515       | 0.6655    | 4.97245    |0.3397        |0.6128   |5.0463    |
>
> | Method          |       |         |Style4           |           |         |Style5           |           |
> | :-------------- | :---- | :------------ | :-------- | :-------- | :------------ | :-------- | :-------- |
> |                 |       | CLIP-T(↑)     | DINO(↑)   | asesthetic   | CLIP-T(↑)     | DINO(↑)   | asesthetic   |
> | With Linear     |       | 0.3612    | 0.6493|5.0580 | 0.3467 | 0.5332    | 5.1983  |
> | Without Linear  |       | 0.3578 | 0.6642    |4.9168    | 0.3460   | 0.5682 | 5.0462 |
>
> ### **Q2. For the SDXL Implementation, the authors mention that style injection is important for the last layers, but injection into all layers compromises text alignment. Is there a principled way for deciding which layers, or some motivating ablations around it?**
>
> A2. We appreciate the reviewer's constructive comment. Our layer selection strategy prioritizes deeper near-output blocks. It is motivated by the ablation studies in [c], which has demonstrated that diffusion models generate outlines primarily in lower-resolution blocks and textures in higher-resolution blocks, respectively. Injecting features across all layers causes noticeable content leakage. To balance style transfer with generation quality, we reduce the injection frequency into lower-resolution blocks. We compare the results between M3S (SDXL) and full-layer injection under identical settings in the Table below. Full injection significantly reduces CLIP-T and aesthetic scores, while slightly increasing the DINO score due to content leakage – a similar observation to that in Q1.
>
> | Method          |       |         |Style1           |           |         |Style2           |           |         |Style3           |           |
> | :-------------- | :---- | :------------ | :-------- | :-------- | :------------ | :-------- | :-------- | :------------ | :-------- | :-------- |
> |                 |       | CLIP-T(↑)     | DINO(↑)   | asesthetic    | CLIP-T(↑)     | DINO(↑)   | asesthetic    | CLIP-T(↑)     | DINO(↑)   | asesthetic    |
> | M3S (SDXL)     |       | 0.3607    | 0.6545    | 4.9802 | 0.3556    | 0.6531    | 5.0581   |0.3422 | 0.6041    |5.1605|
> |Full-layer  |       | 0.3462  | 0.6697  |4.7033   | 0.3351   | 0.7092  | 4.7849  |0.3243  |  0.6567  |  4.9101    |
>
> | Method          |       |         |Style4           |           |         |Style5           |           |
> | :-------------- | :---- | :------------ | :-------- | :-------- | :------------ | :-------- | :-------- |
> |                 |       | CLIP-T(↑)     | DINO(↑)   | asesthetic   | CLIP-T(↑)     | DINO(↑)   | asesthetic   |
> | M3S (SDXL)    |       | 0.3612    | 0.6493|5.0580 | 0.3467 | 0.5332    | 5.1983  |
> | Full-layer  |       | 0.3433  |0.6257   | 4.7400   | 0.3348 | 0.5381 | 5.0002 |
>
>
>
> ### **Q3. The proposed technique is novel and well-motivated for the application domain (which is somewhat narrow, as some of the assumptions made by the authors rely, for example, on images being grayscale).**
>
> A3. We sincerely thank you for recognizing our work's contributions and noting its limitations. In future work, we will validate our multi-style interpolation framework in the complex RGB space and refine the Joint AdaIN module.
>
> [a] Patashnik, Or, et al. "Localizing object-level shape variations with text-to-image diffusion models." ICCV 2023.
>
> [b] Alaluf, Yuval, et al. "Cross-image attention for zero-shot appearance transfer." SIGGRAPH 2024.
>
> [c] Cao, Mingdeng, et al. "Masactrl: Tuning-free mutual self-attention control for consistent image synthesis and editing." ICCV 2023.

---

### Note · Authors · 2025-08-12

# Author Final Remarks

We thank all reviewers for their constructive feedback. We are encouraged that Reviewers **58qS**, **cQmr**, **W249**, and **kzEp** find our framework well-motivated, clearly written, and effective for **training-free multi-style sketch synthesis**. Key strengths noted include its **flexible style blending**, **balanced style-text-quality trade-off**, and **comprehensive experiments**. We especially thank **W249** and **kzEp** for raising their ratings after our rebuttal, recognizing the value of our user study, additional evaluations, and clarifications.

Below is a summary of our responses:

- **To Reviewer 58qS:** We explained linear scheduling of style guidance with supporting results, clarified principled layer selection for SDXL style injection, and discussed extending beyond grayscale sketches.
- **To Reviewer cQmr:** We analyzed object-specific bias (e.g., clown case) via feature injection, AdaIN, and guidance scaling; validated robustness under large style/quality gaps (QuickDraw vs. professional); discussed hyperparameter tuning with default settings for usability.
- **To Reviewer W249:** We added a 44-response user study confirming superior human preference, quantitative evaluation for diverse multi-style pairs, clearer Eq. 5, and main-text failure cases with mitigation; added missing references and clarified novelty.
- **To Reviewer kzEp:** We provided generation time breakdown, clarified baseline categorization, presented user study confirmation, and justified the need for a specialized sketch framework over training-based alternatives.

Please review our detailed responses to each point raised. We hope that our revisions and clarifications satisfactorily address the concerns raised. We sincerely thank all reviewers again for their time, insightful comments, and constructive suggestions.

---

### Decision · Program_Chairs · 2025-09-17

**Decision:**

Accept (poster)

**Comment:**

This paper was reviewed by four experts in the field. Most concerns are solved in the rebuttal and discussions. The paper finally received positive reviews with 2 Borderline Accepts and 2 Accepts.

This paper proposes a training-free style transfer framework for sketch generation. It injects style features by modifying the K and V components in the attention layers, and uses AdaIN to interpolate between different sketch styles.

UniTransfer, a framework controllable video concept transfer. It spatially decomposes videos into foreground, background, and motion flow, using a dual-to-single-stream DiT architecture for fine-grained control. It employs a chain-of-prompt strategy for timestep decomposition, leveraging LLMs to guide the denoising process progressively. It proposes a  dual guidance formulation to disentange the text (content) and style impact.

**strengths**
- clear motivation, well written
- a training-free simple yet effective framework, applicable to both SD 1.5 and SD XL
- good performence quantitatively and qualitatively

**weaknesses**
- insufficient experiments: missing user study, lack of evaluation on diverse styles, missing comparison on running time
- limited technical novelty considering K/V modification, AdaIN are well established methods.

According to the reviews, the reviewers acknowledge the paper's good results and simple idea. The missing experiments are provided in the rebuttal. Following the rebuttal, all reviewers have provided positive ratings for the paper. The AC agrees with the Reviewer W249 that "The proposed method builds on many existing ideas in the literaturee, which could be seen as a limited technical novelty." but "if the model clearly outperforms previous baselines, I would consider this a minor issue".
In the revision, the authors are suggested to incorporate the reviewers' feedback, to include the required experimental results.